# Dinochromosome Heterotermini with Telosomal Anchorages

**DOI:** 10.3390/ijms252011312

**Published:** 2024-10-21

**Authors:** Alvin Chun Man Kwok, Kosmo Ting Hin Yan, Shaoping Wen, Shiyong Sun, Chongping Li, Joseph Tin Yum Wong

**Affiliations:** Division of Life Science, The Hong Kong University of Science and Technology, Clear Water Bay, Kowloon, Hong Kong SAR, China; alvink@ust.hk (A.C.M.K.); kosmo@connect.ust.hk (K.T.H.Y.); wensprussell@gmail.com (S.W.); shysun@swust.edu.cn (S.S.); cliaq@connect.ust.hk (C.L.)

**Keywords:** birefringent chromosomes, chromosome movement, cell cycle, dinoflagellates

## Abstract

Dinoflagellate birefringent chromosomes (BfCs) contain some of the largest known genomes, yet they lack typical nucleosomal micrococcal-nuclease protection patterns despite containing variant core histones. One BfC end interacts with extranuclear mitotic microtubules at the nuclear envelope (NE), which remains intact throughout the cell cycle. Ultrastructural studies, polarized light and fluorescence microscopy, and micrococcal nuclease-resistant profiles (MNRPs) revealed that NE-associated chromosome ends persisted post-mitosis. Histone H3K9me3 inhibition caused S-G_2_ delay in synchronous cells, without any effects at G_1_. Differential labeling and nuclear envelope swelling upon decompaction indicate an extension of the inner compartment into telosomal anchorages (TAs). Additionally, limited effects of low-concentration sirtinol on bulk BfCs, coupled with distinct mobility patterns in MNase-digested and psoralen-crosslinked nuclei observed on 2D gels, suggest that telomeric nucleosomes (TNs) are the primary histone structures. The absence of a nucleosomal ladder with cDNA probes, the presence of histone H2A and telomere-enriched H3.3 variants, along with the immuno-localization of H3 variants mainly at the NE further reinforce telomeric regions as the main nucleosomal domains. Cumulative biochemical and molecular analyses suggest that telomeric repeats constitute the major octameric MNRPs that provision chromosomal anchorage at the NE.

## 1. Introduction

Dinoflagellates are a diverse and ecologically significant group of unicellular eukaryotic protists within the Alveolata clade, which also includes ciliates and apicomplexans [1]. With approximately 2500 extant species spanning around 300 genera [2], dinoflagellates play crucial roles as primary marine producers, contributors to harmful algal blooms (HABs), and essential symbionts of reef-building corals. Dinoflagellates exhibit some of the most unusual and complex chromosome and genome structures known among eukaryotes [3]. One of the most remarkable features of dinoflagellates is their exceptionally large genomes, ranging from 1.5 to 250 gigabases [4,5,6], which are housed within birefringent quasi-condensed chromosomes (BfCs) [7] that lack the canonical histone nucleosomal architecture [8,9,10,11,12,13,14,15,16]. This unique genome architecture is further marked by unidirectionally arranged genes, often presented in tandem repeats [17,18,19], and a substantial proportion of transcriptionally active DNA organized into peripheral loops [20]. Additionally, dinoflagellate chromosomes remain permanently condensed throughout the cell cycle, maintaining a highly organized, cholesteric liquid crystalline state with a constant left-handed twist [7,21].

Telomeres play a critical role in safeguarding chromosome ends, shielding them from being misinterpreted as DNA breaks and preventing deleterious chromosome fusions [22,23]. Telomeric DNA comprises variable-length sequences with conserved minisatellite repeats, spanning from over 100 kbp to several hundred base pairs [23,24,25]. Historically, the structure of dinoflagellate chromosomes has been debated, with earlier models proposing a circular architecture [26]. However, contemporary evidence, including telomerase activity demonstrated via the telomeric repeats amplification protocol (TRAP) assays [27], and the successful application of peptide nucleic acid (PNA) probes in in situ hybridization experiments, has confirmed that dinoflagellate chromosomes are linear and capped with plant telomeric sequences (TTTAGGG)_n_ [27,28]. It is important to note that this plant-like telomere repeat sequence has been consistently found in evolutionarily distant dinoflagellate species (which also vary in their chromosome size and DNA content), including athecate species like *Karlodinium veneficum* [29], *Karenia brevis* [29], and *Amphidinium carterae* [28], and thecate species like *Alexandrium minutum* [30] and *Prorocentrum micans* [28], as evidenced by both genomic [30,31] and FISH (fluorescence in situ hybridization) data [27,28,29,32,33]. While the sequence itself appears to be identical across studied dinoflagellate species, the length of the telomeric repeats may vary. Notably, pulse-field gel electrophoresis of the Bal 31-digested *Karenia brevis* nuclei has shown that these telomeric DNA lengths are longer than those commonly observed for other protists [27]. However, the exact extent of length variation among different dinoflagellate species remains to be fully elucidated.

The dynamic organization of chromosomes and chromatin structure allows for regulated accessibility of DNA, which is crucial for processes such as transcription, genome duplication, and chromosome segregation. During dinomitosis, chromosome ends are positioned within the nuclear envelope (NE), with extranuclear spindles traversing NE tunnels [34,35]. These connections are not only structurally significant, but also have profound implications for the spatial organization of chromosome territories, impacting gene expression and chromosomal behavior [36]. Despite the ecological significance of dinoflagellates in algal blooms and coral bleaching [8], our understanding of how BfCs are positioned within the mitotic NE remains limited. This is particularly crucial given that the NE does not break down during the dinoflagellate cell cycle. The interaction of chromosome ends with the NE forms a critical operational axis, reminiscent of the meiotic “bouquet” and the interphase Rabl configuration observed in various cell types [37,38].

Previous in-gel restriction enzyme digestion experiments demonstrated the release of gene-encoding domains [39], which resided in the outer compartment of dinochromosomes, known as the peripheral chromosomal loops (PCLs). This finding complements the concept of inner “structural DNA” [40], suggesting a complex, multi-compartmental organization of dinochromosomes. The inner–outer compartmentation of dinoflagellate chromosomes exhibits some conceptual parallels with the heterochromatin–euchromatin distinction observed in other eukaryotes [41]. However, the organization in dinoflagellates is unique: the outer compartment (peripheral chromosomal loops or PCLs) contains the only actively transcribed genes, while the inner core likely contains no transcriptionally active domains and is termed structural DNA, effectively separating the coding and non-coding sequences. This arrangement differs from those of typical eukaryotes, where euchromatin and heterochromatin are often interspersed with both coding and non-coding sequences throughout the chromosome. In dinoflagellates, this clear spatial separation, with transcriptionally active regions exclusively in the outer compartment, represents a distinct chromosomal architecture. This unique division of labor likely employs subcompartmentation for differential gene expression, facilitating the multiple life-cycle stage transitions characteristic of dinoflagellates.

Telomeres, located at the ends of chromosomes in eukaryotes, consist of repetitive non-coding DNA sequences. In both plant and mammalian cells, special mechanisms are installed to prevent telomeres from progressively shortening during each round of DNA replication, as the 5′ end of the lagging strand cannot be synthesized after the removal of the last RNA primer, resulting in a 3′ overhang [42,43,44]. Telomeres and telomeric nucleosomes (TNs) are widely acknowledged as vital protective structures that safeguard the integrity of chromosome ends during replication [44,45]. However, their potential roles in other cellular processes have been less addressed.

Although canonical histone complements are significantly reduced in dinoflagellate genomes, histone variant presence suggests essential roles in chromosomal transactions that are perhaps commonly obscured by canonical architectural nucleosomes in other organisms. In both plant and mammalian systems, histone H3 at lysine 9 methylation (including H3K9me3 and H3K9me2) is a critical epigenetic marker that influences the compaction state of telomeric nucleosomes and rRNA loci [46,47,48,49,50]. Its interaction with the Su(var)3–9 histone H3–K9 methyltransferase (Suv39H, plant homolog NtSET1, and SUVH4/KYP) stabilizes telomeric nucleosomes, with Suv39H null mutants showing decreased H3K9me3 levels and abnormal telomere elongation, demonstrating the importance of H3K9me3 in maintaining telomeric integrity [51,52,53,54]. The retention of telomeric nucleosomes (TNs) in dinoflagellates, despite the absence of canonical architectural nucleosomes, implicates their potential roles in orchestrating system-level chromosomal organization in these unique protists.

The selected complement of histone variants likely performs ‘system-level’ functions, despite forming complete H2A:H2B||H3:H4 nucleosomal pairs [55]. Standard micrococcal nuclease digestion of BfCs failed to generate a typical ‘nucleosome ladder’ [56], aligning with the proposal that most BfC modules adopt superhelical–plectonemic conformations [20,40]. Several models have been proposed to elucidate the intricate organization of dinoflagellate chromosomes. Notable among these are the “toroidal chromonema” model [14], which envisions chromosomes as toroidal bundles of DNA strands based on circular chromosome organization, and the “stacks-of-DNA-discs” (or cholesteric liquid crystal) model [7,57,58], which proposes a linear chromosome organization with each chromosome arranged as layered discs of nested DNA arches. Previous models of BfCs, based on purported higher-order coiling, likely represented partially decompacted chromosomes. It would not be conceptually helpful to evoke ‘liquid crystalline chromosomes’ to describe the outer higher-order structure, as inter-chromosome interactions would not have been discrete. Comparative TEM studies by the pioneer TEM developer Kellenberger group demonstrated that low-protein nucleic acids (unlike nucleosomal nucleic acid) exhibited apparently artifactual cholesteric ‘DNAplasm’ [59,60,61]. Furthermore, recent studies suggest that the surface peripheral chromosome loop domains are transcription-mediated and will not be in liquid crystalline form [62]. Given these considerations, we prefer to refer to the readout description of ‘Birefringent Chromosomes’. This observation could result from either the majority of nucleosomal compartments being less accessible, or the dominant non-nucleosomal superhelicity disrupting the helical constraint of genomic DNAs. These findings suggest that the chromatin structure in BfCs deviates significantly from the conventional nucleosomal organization, potentially impacting DNA accessibility and genome function. Dinomitosis is conducted without an intranuclear spindle and occurs without nuclear envelope breakdown or openings. Despite the loss of canonical core histones as the primary chromosomal architectural proteins, dinoflagellate genomes have retained histone variants whose functions have not yet been fully elucidated [55]. In the current study, the comparative analysis of Micrococcal nuclease-resistant profiles (MNRPs) and DNase I resistance patterns suggested periodic structures resembling telomeric nucleosomes. Using biochemical and immunological approaches, we examined core histone distribution and inter-histone interactions, particularly in relation to their cross-linking with supercoiled DNA. Moreover, our biochemical and immunological findings indicate that telomeric nucleosomes constitute the major fraction of nuclear nucleosomes. Our findings suggest one end of each BfC being anchored within the NE, displaying a high degree of compaction consistent with telomeric nucleosomes, while the other end is iso-connected to the nucleoli. Importantly, this relative positioning persists throughout G_1_ with profound implications for chromosomal dynamics and gene expression with nucleolar activity.

## 2. Results

Immunoblot analysis of extracted nuclear proteins revealed an unexpected distribution: *Crypthecodinium cohnii* histone H2B (CcH2B) was predominantly in the cell pellet, while dinoflagellate histone-like proteins (dHlps) and α-tubulin were mainly in the supernatant (Appendix A). This distinct distribution suggests that core histones and dHlp occupy separate compartments, unlike their typical co-localization on chromosomes in other eukaryotes.

Time-lapse birefringence microscopy revealed stable G_1_ chromosome configurations, with one end anchored to the nuclear envelope and the other extending into the nucleolar region (Appendix A), supporting our previous findings regarding the inner and outer compartments within BfCs [10,63]. Together, these results indicate that dinoflagellates possess a unique nuclear organization in which core histones and dHlp reside in separate compartments. This arrangement is likely linked to the stable chromosome structure observed throughout the cell cycle and has significant implications for chromosome dynamics and nucleic acid isolation methods in these organisms.

### 2.1. Isolated Chromosomes Have Two Ends

Fluorescence microscopy of isolated *Karenia brevis* chromosomes revealed a distinct difference between the two chromosome ends compared with in-nuclei observations: one end appeared more decompacted, while the other remained more condensed (Figure 1A,B). Isolated BfCs often exhibited a stage-2 cation-chelation-mediated karyomorphology [64], with a stretched, screw-like structure encircling a central core. This consistent spiraling suggests anchorage-dependent decompaction with higher-order de-coiling, unlike the expected erratic patterns without anchorage.

DNA dyes like DAPI, which primarily visualize condensed DNA, revealed bright fluorescent structures representing inner chromosome compartments that remained partially condensed after chelation or processing [64]. In contrast, peripheral chromosome loops (PCLs) [40], appeared as non-fluorescent, birefringent voids, creating ‘apparent gaps’ between chromosomes.

DAPI staining of isolated nuclei revealed well-separated, round-shaped BfCs, indicating that fixation resulted in vertical compaction, which in turn appeared to push the BfCs toward the periphery, accompanied by concurrent decondensation (Figure 1B, red arrows). This contrasts with the elongated shape of BfCs observed *in vivo* and in isolated chromosomes, suggesting that decondensation occurs at the chromosomal ends, rather than at the nucleolar ends. Notably, several isolated chromosomes lacked a condensed domain at either end (Figure 1B, green arrows).

Following micrococcal nuclease digestion, most BfCs were digested, leaving behind residual dots (Figure 1C, red arrows). A smaller fraction of BfCs exhibited poor DAPI staining, appearing green in color. These chromosomes were longer and occasionally featured joined sister BfCs (Figure 1C, orange arrows), likely representing G_2_ phase BfCs requiring duplication prior to segregation.

Transmission electron microscopy (TEM) photomicrographs (Figure 1D) and observations from time-lapse videography (Appendix A) revealed the immobility of BfCs and the presence of nuclear envelope insertions. These insertions likely represent telomeric nucleosome anchorages, providing a physical link between chromosome ends and the NE.

A stringent test of telomeric anchorage would involve BfC decompaction–recompaction [64] by employing a sub-decondensation concentration of EDTA. The metripol system a compact karyobirefringent type with the standard 360-degree changes in false color codes, confirmed the structural uniformity between BfCs and the intra-karyogenomic organization within each BfC, as well as each focused layer having chromosomes on the same optical plane (Figure 2A). When focused on the nuclear surface, distinct ‘black’ labels were observed at BfCs located at the equatorial region (green and yellow), suggesting that chromosome ends within the NE had different compositions compared with the rest of the BfC. The mirror image, along with its absence at the other 90 degrees, was indicative of a birefringent void. Cation chelation by EDTA led to the planned BfC partial decompaction (Figure 2B), resulting in the loss of orientation order. Notably, the chromosome ends remained, but were now surrounded by a birefringent void (black), supporting the EDTA-mediated open-end decompaction, while each BfC largely maintained its anchored position. The black arrow in Figure 2B points to one chromosome end expulsed from the NE. We opined that these were attributed to varying molecular activities (e.g., transcription) across different BfCs. Despite retaining some birefringence, the decompaction–re-compaction displaced many BfCs out of the focal plane, indicating that the normal compaction was at a tightly organized karyogenomic equilibrium with interchromosomal space. Given the isolation severity of the BfCs, it was likely that a segment of the telomeric enclave was coerced to detach. The severed telomeric fragments could have contributed to the non-chromosome end signals [27] observed with in situ hybridization.

### 2.2. Restriction Enzyme Pre-Digestion Gave Different Micrococcal Nuclease Resistance Pattern (MNRP)

We found that the freeze-thawing (fracture) technique was more efficient at extracting histones than methods involving sonication and grinding in liquid nitrogen (Appendix A), suggesting that core histones in dinoflagellates are not freely extractable but likely associated with membrane compartments that were cracked by freeze–thawing.

The standard comparative method for assessing chromosome nucleosomal compaction is the micrococcal nuclease resilience pattern, typically based on unbound/unattached chromosomes. We compared the micrococcal nuclease resilience patterns (MNPPs) between the isolated chromosomes and in-nuclei digestion, taking advantage of the large *Karenia* nuclei (~10 µm) that did not decompact in the nuclei isolation buffer and remained birefringent [64].

The chromosomes isolated from nuclei exhibited distinct MNRPs, with a noticeable decrease in resistance time points for both ribosomal DNA (rDNA) and telomeric regions (Figure 2C–F), suggesting the absence of major protected domains in the isolated chromosomes. This supports our hypothesis that the nuclear envelope (NE) end and the rDNA ends were severed during processing. If both chromosomal termini were unprotected, telosomal nucleosomes (which are labile) would not provision micrococcal nuclease protection for 24 h.

The isolated chromosomes did not exhibit the prominent MNase-protected domains that the whole nuclei did (red boxes, Figure 2D compared to Figure 2E), with apparent m.w. lower than 100 bp, and the protected domain remained largely undigested, even after 24 h (compared to digested nuclei in Figure 2E, green box region). The MNRP Southern blot (Figure 2C–F) suggested that this fraction was neither telomeric nor contained rDNA. The highly protected fraction (HPF), retained within the nucleus and ranging between 0.5 to 12 kbp (Figure 2E, highlighted by the green box), showed only a weak Southern blot signal with rDNA probes, indicating the presence of major resilient structures, in addition to the non-HPF-rDNA loci. This indicated that at least some of the rRNA loci were protected with nucleosomal domains.

Only a minor fraction of the micrococcal nuclease-resistant (MNR) chromosomes was detected after 24 h of digestion with isolated chromosomes as a substrate (Figure 2D, red box). However, a significantly larger amount of the MNR fraction was observed in the whole nuclei (Figure 2E, red and green boxes), suggesting that most of the MNPP-containing fraction was found within the NE or the permanent nucleoli, rather than within the isolated chromosomes. This observation corresponded to the fluorescent photomicrograph of BfCs (Figure 1B) and suggested that they were physically severed from their anchorage within the NE during nucleic acid preparations and chromosome isolation procedures.

### 2.3. MNase Resistant Patterns (MNRPs) with Telomeric Southern Blot Analysis

Restriction enzymes preferentially release gene-encoding domains, suggesting easier separation of the PCL outer compartment [39] from resilient inner compartments [20]. Our study showed quicker elimination of middle segments by restriction enzymes, indicating the release of a potential resilience domain and faster access to telomeric anchorage. This aligns with previous TRAP assays showing increased accessibility of chromosome ends [27], suggesting a connection between telomeric regions and inner compartments. Prior Bal31 nuclease digestion released unexpectedly large (50–70 kbp) telomeric sequences [27], suggesting inaccessibility of the nuclear envelope (NE) during normal digestion. Assuming NE insertion, restriction enzyme cutting would expose inner compartments to MNase, theoretically resulting in a faster MNRP.

In telomeric Southern blots without restriction enzyme pre-digestion (Figure 3A), two distinct resistance fractions with high and low molecular weight (m.w.) were observed, compared with only one major resistant fraction (low m.w.) in MNase digestion-only blots, implying that pre-digestion released a high m.w. fraction and increased MNase susceptibility (Figure 3B). Interestingly, pre-digestion with restriction enzymes did not result in the expected decrease in resistance, but rather an increase in the last resistant fraction (Figure 3C,D,F), suggesting a link between the inner domain and the telomeric region, with restriction enzymes enhancing MNase accessibility to these domains.

To further validate the MNRP approach and investigate the role of RNAs in telomeric complexes, we modulated the assay using DNase I, which does not digest RNAs and does not require Ca^2+^ for its activity. The deproteination of the DNase I-digested sample resulted in a nucleosome-like ladder pattern (green box) on the telomeric Southern blot (Figure 3E). The apparent distance between inter-ladder bands was slightly greater than the expected 160–200 bp for nucleosomes composed solely of DNA. The distinct differences between the DNase I and MNase resilience patterns support the existence of a fraction ranging from 5–20 kbp (with a persistent 7 kbp fraction after 17 h) that could represent the shelterin complex or repeats containing the telomeric elements. The slight fuzziness observed may be attributable to associations with the NE membrane (Appendix A). Proteinase treatment of MNase or DNase I-resistant fractions revealed specific ladder-like patterns, indicating that proteins were the major macromolecules contributing to the nuclease resistance, while RNAs likely contributed to the rest of the protective binding interactions.

The reproducibility of MNRPs reflects a consistent chromosome architecture and NE orientation, as evidenced by the rainbow color change indicating a shift in chromosome orientation (Figure 2A), with one chromosome end anchored to the NE and the other pointing toward the nucleus center. Unresolved streaks in the MNRPs would imply selective NE attachment of chromosomes, which was not the case (Figure 3B–D). The digestion pattern showed a low-m.w. resistant fraction emerging after 2–2.5 h of MNase digestion, coinciding with the appearance of the major telomeric resistance fraction (Figure 2D). Following MNase digestion of nuclei, most BfCs were digested, leaving behind identifiable remnants (red pointers in Figure 1C).

The higher resolution from restriction enzyme pre-digestion suggested that the MNase-resistant signal was due to physical BfC continuity to the NE (Figure 3) that sustained variation attributed to differential severances. Staining with the lipid stain Patman revealed concentrated dots, likely the telomeric enclaves, surrounded by the blue fluorescent membrane lipids (Appendix A). The remnants of MNase-resistant domains, when separated by sedimentation, were positively labeled with the DNA stain SYTOX green, as well as with the lipid dye Patman (Appendix A), further supporting the notion that telomeric anchorages are contained within the NE.

### 2.4. The Non-Nuclear Envelope Chromosome Ends Contain the rDNA Loci-NOR That Contributed to the Nucleoli

Southern blot analysis using rDNA probes revealed a high-molecular-weight MNase-resistant fraction persisting through the initial time points (Figure 2F, red squares). This distribution suggests synchronized resistance across multiple chromosomes carrying rDNA loci. This high-m.w. fraction decreased with increasing digestion time, showing resilience prior to 24 h, suggesting that a substantial number of chromosome ends were associated with rDNAs. The lack of association between rDNA positive and telomeric repeat signals (lower m.w.) at extended digestion times suggested that rDNAs were absent at the NE end. Previous TEM studies have indicated that nucleoli consist of decondensed chromosomal loci [66]. Metripol and birefringence microscopy suggested that the non-NE ends were ‘opened’ into the nuclear center where the nucleoli were located, indicating an available domain, likely nucleosomal.

Dinoflagellate nucleoli persist throughout the cell cycle, as demonstrated in *Crypthecodinium cohnii*, due to nucleolus organizer region (NOR)-rDNA loci at chromosome ends [66,67]. This persistence, along with B-end chromosome ends associating with nucleoli, suggests a nucleoli–NE axis in chromosomes. Interestingly, rDNA showed a different resistance pattern than telomeric repeats (Figure 2E,F), implying that some open-end rDNAs were in a protected conformation resistant to MNase digestion for up to 5 h. The gradual decrease in the high m.w. fraction (Figure 2F, red box) over 24 h indicated a substantial number of chromosome ends being associated with rDNAs. The high-m.w. rDNA-positive bands were notably absent from the MNRPs of isolated chromosomes, supporting the hypothesis that the B termini (referring to the non-NE end) were cleaved during chromosome isolation, with the open end being the sole location(s) of rDNA repeats. The concentrated domains (higher m.w.) suggested that rDNA repeats were organized in the NOR domain, which, in other cells [41], helps to shape chromosomes. Moreover, the lack of MNR signals during the median digestion timepoints suggests that non-telomeric nucleosome-like domains were mostly absent within the main BfCs architecture (Figure 3E).

In addition to the high- and low-m.w. rDNA fractions, we observed streaks reacting to rDNA probes, suggesting lower resilience binding. This indicates that other nucleolar macromolecular complexes [68], such as ribosomal nuclear particles (RNPs) [69] and small-nuclear RNPs (snRNPs) [70], may contribute to the protected conformation in dinoflagellates. Following MNase digestion, the typical nucleolar ‘voids’ were no longer visible (Figure 1C). Consistently, remnants exhibited green fluorescence, rather than the usual light-blue DAPI staining, hinting at the presence of partially digested ribosomal DNA:RNA complexes from the nucleoli.

### 2.5. Marker Repetitive Elements Differentially Labeled Micrococcal Nuclease Profiles

The analysis of continuous recordings indicated that the positioning of the chromosomes remained during the G_1_ phase (Appendix A), as evidenced by the lack of change, even with cation–chelation-mediated partial decompaction–recompaction events (Figure 2A,B). This stability suggests a stable karyogenomic arrangement of the chromosomes in relation to both the outer and inner compartments of the BfCs, corroborating their relative positioning and the extension of the ‘non-NE’ end toward the nucleoli, as captured in the dynamic continuous recordings.

Given the relative stability of the telomeric enclaves, we inferred that this would ensure a stable spatial karyogenomic architecture in relation to both the outer and inner compartments of the BfCs. Assuming the karyogenomic organization of the inner compartment and more accessible outer compartments comprising the PCLs, Southern blots with coding sequences (complementary DNAs) would be expected to reveal these regions at early time points in the MNRP. Repetitive elements, including those associated with rDNAs, play a significant role in genomic evolution and organization. Previous studies have shown that rDNAs, such as 5S rDNA, have evolutionary mobility as mobile elements [71,72,73]. This distribution of repetitive elements, including those associated with rDNAs, often marks major evolutionary events and contributes to genomic diversity. We conducted comparative Southern blot analyses using probes for known repetitive elements and complementary DNAs. Utilizing known repetitive elements (cc20 and cc18) previously identified within gene-encoding domains [74] contributed to delineating the phylogenetic relationships within the Crypthecodiniaceae family [75]. Notably, cc18 and cc20 gave distinct patterns of MNRPs, despite not being part of the highly MNase-resistant domains (Figure 3G,H). When comparing the Southern blot signals of cc20 and cc18, we observed that the cc20 signal extended further in samples subjected to more extensive digestion (Figure 3G,H), thus confirming the presence of different surface compartments.

When using cDNA as a probe, a markedly streaky variable MNRP was observed, whereas the actin gene probe resulted in a weaker MNRP signal that could only be observed with extended incubation (Appendix A). The strongest signal occurred at the early digestion stage (10 units, Figure 3B), indicating that highly expressed transcripts are preferentially located in more accessible domains. The actin Southern blot signal was only detected at median digestion points, but not at the initial or final time points (Appendix A), with signals only detectable after extended exposure (over 1 h), suggesting that actin genes, which are encoded in tandem repeats, are likely located at more distal BfC ends.

The cDNA positive signals, though weaker and more variable than those of the repetitive elements, were supportive of selective gene-coding domains in areas closer to chromosome ends. This is consistent with the prior proposal that actively transcribed genes are located with unidirectional tandem repeats, whereas lesser expressed genes were in the single-copy form [18]. Our data indicated that the lesser expressed genes could be more cortically located and associated with histone–nucleosomal regulation.

Considering the higher molecular crowding with increasing gel unit (or digestion time), the longer streak toward the end indicated reduced accessibility of loci closer to the space between BfCs. Cumulatively, our observations with cDNAs and specific markers corresponded to expected karyogenomic positioning, rendering MNRPs a valuable tool for investigating telomeric–nucleolar dynamics within the broader context of chromosomal dynamics. This is particularly the case as the inner compartment lacks Ca^2+^ [76], which is required for MNase activity, implicating slower rates of digestion per outer–inner compartmental disassembly.

### 2.6. Sirtinol, Non-Specific H3K9me3 Inhibitor Led to Concentration-Dependent Chromosome Decompaction

Telomeric nucleosomes were commonly enriched with histone H3 variant H3.3. Comparative sequence analysis of six *Crypthecodinium cohnii* histone H3 variants (UHA57726.1, UHA57727.1, UHA57728.1, UHA57729.1, UHA57730.1, and UHA57731.1) with the human H3.3 protein has uncovered notable distinctions (Appendix A). Histone H3.3 is a variant of H3, differing from the canonical H3 by only a few amino acids in evolutionarily distinct organisms, including humans, *Drosophila, Xenopus*, and *Arabidopsis* [77,78]. Multiple sequence alignment of *C. cohnii* H3 variants with canonical H3 proteins from other organisms, including apicomplexans, perkinsids, and diatoms (the latter two having H3 but not H3.3), revealed that *C. cohnii* H3 variants exhibited unique features distinct from both their close relatives and more distant eukaryotes (Appendix A). *C. cohnii* H3 variants show regions with highly variable sequences compared with apicomplexans and possess extra sequences in both the N-terminal and histone fold domains not found in other eukaryotes. Moreover, *C. cohnii* H3 variants do not share the “(A/Q/E/T/H)A(I/L/V)(L/G)” sequence motif conserved in many eukaryotic H3.3 proteins, nor do they possess the “SAV(M/L/A)” motif characteristic of canonical H3 in many organisms. Instead, these variants display a diverse “(Q/E/S)(A/G)(I/L)(L/S/E)” motif, which more closely resembles the H3.3 “(A/Q/E/T/H)A(I/L/V)(L/G)” motif (Appendix A). Despite these sequence differences, both CcH2A and CcH3 variants were still recognizable by the commercially available H3 and H2A antibodies used in this study (Appendix A), which target the highly conserved histone fold region shared by canonical histones and histone variants. Furthermore, most (five out of six) of these *C. cohnii* variants exhibited similar transcription levels in G_1_ and S-G_2_ cells (data from in-house cell cycle transcriptome), reminiscent of the expression pattern of H3.3 in other eukaryotes [79,80,81]. The post-translational modifications predictions indicate potential methylation at the H3K9 position on three of the *C. cohnii* H3 variants (UHA57726.1, UHA57729.1, and UHA57731.1), but not on other lysine residues, such as H3K56 or H3K27, which are known modification sites in other contexts [82,83]. These reduced epigenetic marks likely evolved with the loss of canonical core histones, and single-residue modification might be enough to distinguish the already differential chromosomal landscape.

The interplay between histone H3 lysine 9 trimethylation (H3K9me3) and acetylation (H3K9Ac) represents a critical epigenetic switch that governs chromatin state and gene expression [84,85]. These two modifications are mutually exclusive on the same lysine residue and exhibit a reciprocal inhibitory relationship, where each modification can suppress the establishment of the other [86,87], often associated with opposing chromatin states [88]. H3K9Me3 is typically linked to heterochromatin formation and gene silencing, while H3K9Ac is associated with (open and accessible) euchromatin and active gene transcription [89]. Sirtinol, a sirtuin deacetylase inhibitor [90], was employed to investigate the role of histone deacetylation in dinoflagellate chromatin compaction and NE integrity. In various eukaryotic systems, including plant and mammalian cells, sirtinol indirectly promotes H3K9 acetylation, a modification associated with active transcription and open chromatin states [86,91]. While sirtinol does not directly alter histone methylation, its enhancement of acetylation can indirectly reduce methylation at the same residue. Interestingly, these epigenetic modifications may also influence telomere integrity, which is particularly relevant in the context of ROS (reactive oxygen species) imbalance. Telomeric DNA is especially reactive with ROS and prone to oxidative damage [92], partly due to the preferential binding of Fe^2+^ to telomere repeats [93]. This susceptibility to ROS-induced damage highlights the importance of proper chromatin structure in protecting telomeric regions.

Immunolabeling with anti-H3K9me3 antibodies revealed robust labeling of the swollen nuclear envelope (NE) (Figure 4A). Complementary immunoblot analysis of the H3K9me3 epigenetic mark in sirtinol-treated *Karenia brevis* cells demonstrated a decrease in H3K9me3 levels following sirtinol treatment (Figure 4B). The sirtinol effect in *C. cohnii* was much slower and more regional (Figure 4C,D). This contrasts with AMSA (Figure 4E), which dose-dependently caused decompaction of the main body of chromosomes leading to nuclear eruption. We observed that central nucleolar positions became more decondensed prior to the cortical region. This difference in decondensation was evident at 10 and 50 µM concentrations at the 6 h mark (Figure 4C). These findings suggest that sirtinol-induced effects, whether specific or non-specific, affected chromosome higher-order organization at the cortical areas, transmitting to decondensation at the chromosome ends within the NE. Histone acetylation selectively affected chromatin organization near the nucleolus and telomeric regions, thereby influencing NE stability and overall nuclear architecture.

Our experiments also demonstrated dose-dependent restricted BfC decompaction in *C. cohnii*, without significant nuclear decompaction until 50 µM was reached (Figure 4B). Notably, we found that the eventual decompaction of the main body coincided with DNA staining within the nuclear envelope, which was previously unstainable. This observation unequivocally demonstrated the presence of BfC termini within NE DNAs in *C. cohnii*, which we refer to as telosomal enclaves for the purpose of discussion, that were connected to the decompaction of telomeric nucleosomes. Based on our findings for *C. cohnii*, we suggest that, despite the whole chromosome connection, most of the telomeric nucleosome-mediated compaction occurred in the TA direction, with a lesser contribution to the chromosome bulk, which was more associated with topoisomerase.

We examined the roles of H3K9acetyl and H3K9me3 in both synchronous and asynchronous cell populations, exploring their distinct, yet interconnected, functions. To further elucidate the role of histone modifications in cell proliferation and chromatin structure, we treated *Crypthecodinium cohnii* with chaetocin, a selective inhibitor of the lysine-specific histone methyltransferase SUV39H1 responsible for H3K9me3 [94]. Despite the absence of architectural nucleosomes in dinoflagellates, chaetocin significantly impeded cell proliferation (Figure 5A), increasing the doubling time of *Crypthecodinium cohnii* from 8–10 h to roughly 24 h, and decreased H3K9me3 immunoblot signal (Figure 5B), despite these cells lacking a conventional nucleosome structure. In synchronous cells, 20 µM chaetocin specifically delayed the S phase when added at the late G_1_ phase (Figure 5C). The synchronicity of the delay suggested that this epigenetic mark (H3K9me3) plays a critical role in the regulation of the cell cycle at a systemic level. This enhanced detection of H3K9me3 at the NE may be attributed to post-fixation processing, which could displace the NE, making it appear thicker and obscuring the nucleolar region from view.

We performed multiple attempts to immunolabel core histones, including using the specific anti-ccH2A antibody, but obtained inconsistent results. The anti-H3K9me3 immunolabeling appeared to be influenced by the enlarged nuclear envelope (NE), which may increase the propensity for aggregation during spin-down procedures. This suggests that H3K9me3 contributes to chromatin disassembly, potentially explaining chaetocin’s inhibitory effect on S-phase exit. These observations indicate that the H3 epigenetic mark may drive NE enlargement, consistent with reports of a double-layer membrane structure in dinoflagellate NE [95]. Additionally, higher concentrations of sirtinol indicate the potential involvement of redox changes underlying the NE alterations, likely due to its iron-chelating ability *in vitro* [96]. Furthermore, a few immunofluorescent images displayed strong anti-H2A staining (Appendix A), whereas the H3 antibody frequently produced unclear or negative results, despite positive signals in Western blot assays. This discrepancy suggests that the H3 epitope is inaccessible *in vivo*, likely due to the NE’s surrounding environment, which is less exposed compared with peripheral H2A:H2B pairs. As the decompaction of birefringent chromosomes (BfCs) increases DNA dye accessibility, we infer that histone epitopes are generally hidden, with no significant histone domains present in the bulk of the chromosomes.

### 2.7. Supercoil Cross-Linking Required for Exclusion-Mediated Telomeric Nuclesome–Octamer Isolation

Our ongoing data indicated that histone epigenetic modification plays a role in the regulation of chromosome ends, coupled with the dynamic compaction of the BfCs. We reason that the major plectonemic landscape would be detrimental to the minor nucleosomal component and that the fixation of the plectonemic fraction would have facilitated the isolation of the telomeric nucleosomes. Bearing in mind the lesser TN compaction level, additional protein–DNA crosslinking may be required.

Psoralen, a DNA cross-linking agent that preferentially binds to supercoiled DNA, was utilized to enhance the stability of histone-containing nucleosome complexes during chromatin/nucleosome isolation and electrophoretic analyses [97,98]. The crosslinking occurs in linker DNA, whereas the nucleosomal DNA is protected, which allows whether a DNA region had been occupied by a nucleosome or not to be distinguished [98]. Surprisingly, psoralen treatment enhanced the persistence of histone-containing complexes. The combination of crosslinking agents (psoralen + DSG) provided distinctive resilience patterns on one-dimensional gel electrophoresis (Figure 6A–J), supporting our thesis that supercoil cross-linking increased octameric stability for protein–DNA crosslinking, thereby facilitating the isolation and analysis of stable telomeric nucleosome complexes in dinoflagellates. Correspondingly, DAPI-stained photomicrographs of psoralen-treated G_2_ nuclei (blue arrows, Appendix A) exhibited an oval shape compared with the circular G_1_ nuclei and untreated control without fixation (Appendix A). In both cases, an increase in nuclear volume was observed, suggesting that supercoil domains were interspersed unevenly with non-supercoil domains, which exhibited differential shrinkage in response to fixation. The D+P and D+F treatments (last lane) yielded the most electrophoretically resolvable domains with anti-H2A antibody-mediated immunocapture (Figure 7A–C). These treatments resulted in similar lowest molecular weights (26 kDa), suggesting that the captured complex exhibited H2–H3 connectivity in addition to the canonical preferred H2A:H2B and H3:H4 pairs, with predicted m.w. values that were also found in the higher m.w. fractions.

Differential cross-linking between supercoil and non-supercoil domains elicited a condensation–relaxation effect on higher-order chromatin structures. As psoralen has affinity for DNA–DNA supercoil crosslinking, rather than nucleosomal solenoid coiling, these results demonstrated that dinoflagellate nuclei (dinokaryon) primarily contained non-nucleosomal supercoiling. Furthermore, the nuclear envelope became labeled with DAPI (greenish, indicated by green arrows, Appendix A) following psoralen crosslinking, suggesting that chromosomal crosslinking led to the transmission of condensation–relaxation effects into the TA regions, with some differences observed between G_2_ and G_1_ nuclei. The G_2_ BfCs appeared more separated after psoralen treatment (Appendix A), indicating differential condensation. This further demonstrates the presence of TA regions in both G_1_ and G_2_ cells, with distinct responses to psoralen-induced crosslinking.

The immunocapture of nucleosomal histone complexes requires highly stable nucleosomes. This would be a more stringent demonstration of *in vivo* nucleosomal formation, but will require non-histone fold anti-bodies that were not preoccupied with DNA binding. Our anti-H2A antibody will suffice this requirement as it was generated against the conserved N-terminal tail that exhibited higher accessibility.

We propose that prior failed attempts to observe MNRPs [13] can be attributed to telomeric anchorages at the NE, and that fixation-mediated decondensation of the main chromosome body spreads to the rRNA-containing end, concurrently inducing decondensation in this region. However, attempts to conduct immunocapture with the NE fraction did not yield the canonical pattern, and we relate this to sampling incurred decompaction, as well as the possibility that the nucleolar end comprises no or few nucleosomes. To circumvent this problem, we utilized fixation-coupled immunoprecipitation, building on our MNRP investigations. Following cross-linking, MNase-resistant populations were predominantly found in the non-bound supernatant fractions of samples cross-linked with DSG-formaldehyde or DSG-psoralen, but again, the dual agents DSG and formaldehyde were necessary to immunocapture histone-containing complexes that suggest nucleosomal molecular weight (Figure 7D–F). This unequivocally demonstrated the TAs within the NE require pre-restriction of the plectonemic supercoil prior to the DSG-mediated protein–DNA cross-linking.

The presence of nucleosomal and non-nucleosomal supercoils would have resulted in two slopes with different increases in repetitive repeats; the smaller amount of MNase resistance domain (~2.5 kbp apparent m.w) with slower electrophoretic mobility in the first dimension did not exhibit differential mobility in the second dimension (Figure 6H, linear line, blue label).

This interpretation was further supported as DSG–psoralen cross-linking being more effective than DSG–formaldehyde, as the latter combination prefers protein–protein cross-linking that would have incurred mobility shift. Notably, there was an apparent shift in molecular weight (MW) of several kilobases after crosslinking, different from that of mononucleosomic DNA of 100–200 bp (without proteins) to approximately 500 bp of a protein–DNA complex, indicative of an octameric complex. Since nucleosomal octamers should be associated with H2A:H2B and H3:H4 associations [99,100,101], we conducted additional co-immunoprecipitation analysis. Neither anti-H2A nor anti-H3 immunoprecipitants were associated with tubulin, confirming little contamination of the house-keeping proteins. The anti-H2A immunoprecipitation successfully pulled down H2B and H3, further supporting nucleosomal interactions (Figure 7A,B).

Ethidium bromide staining and telomeric Southern blot overlapped only at the higher m.w. subdomain without crosslinking, while psoralen pre-crosslinking resulted in almost complete overlapping (>95%), suggesting that most of the telomeric MNRPs were contributed by nucleosomal domain, and that the telomeric repeats had two coiling densities with different slopes (purple). We interpret the faster-mobility fraction as representing the open ends, which will have a spectrum of different repeats, whereas the higher-molecular-weight fraction represents the NE-associated fraction. The uniform first-dimensional mobility implicated potential crosslinking, as demonstrated by the Patman-labeled MNRP. The complete shift observed with psoralen pre-treatment could also be contributed by mixed telomeric elements at the telomeric ends, as demonstrated with the changed linearity of the telomeric Southern blot (green color). Alternatively, psoralen might destabilize the original domain, leading to MNase digestion. In either case, the telomeric Southern blot suggested that the telomeric nucleosomal domain was the major micrococcal nuclease-resistant domain. Although we cannot unequivocally exclude the presence of low levels of repetitive nucleosomes interspersed among the BfCs, the lack of a nucleosomal-ladder resilience pattern with the cDNA hybridization probes (Appendix A), which would only hybridize with the peripheral chromosomal compartment, aligns with telomeric nucleosomes being the predominant nucleosomal domains.

## 3. Discussion

Chromosome positioning within the nucleus is fundamental to several critical cellular processes, including replication, segregation, differential gene expression, and susceptibility to DNA damage. This is true regardless of whether chromosome ends are located at telomeres, nuclear anchorages, or within chromosome territories. In eukaryotes, chromosome ends have the potential to be mistakenly identified as damaged or broken DNA, necessitating their protection from cellular DNA damage response mechanisms [102]. The observed non-random lengths of telomeres [45] and the maintenance of consistent chromosome dimensions (karyotypes) indicate deliberate regulatory processes, rather than random occurrences. The nucleolar location at the non-telomere-associated (non-TA) end also suggests a nucleolar–nuclear envelope (NE) axis, similar to the centromere–telomere axis reported for certain chromosome territories [103] that would relate each chromosome uniformly to the effects of ribosomal synthesis.

Our findings suggest that telomeric nucleosomes are the primary nuclear locations for dinoflagellate core histones. This conclusion is supported by crosslink-dependent immunocapture, the limited effects of sirtinol on the bulk of BfCs at lower concentrations (Figure 4), and the results of two-dimensional gel analyses (Figure 6). Additionally, the presence of MNase-resistant proteins (MNRPs) supports the absence of architectural-type nucleosomes throughout the chromosome bulk, aligning with the non-canonical gene-encoding mechanisms observed in dinoflagellates [18]. This is further corroborated by the lack of specific labeling in immunofluorescence experiments (Appendix A), confirming the minimal presence of traditional nucleosomes in these regions. Furthermore, when separated by sedimentation, remnants of MNase-resistant domains were positively labeled with both the DNA stain SYTOX Green and the lipid dye Patman (Appendix A), supporting the notion that telomeric anchorages are contained within the NE.

We treated cells with chaetocin at a concentration of 20 µM for 24 h following cell cycle synchronization. Chaetocin, known to inhibit SUV39H1, not only reduces H3K9me3 levels [104], but may also disrupt redox-sensitive enzymes [105]. While the potential cytotoxic effects of chaetocin were likely related to alterations in cellular redox state, which likely affect higher-order chromosome structures, we observed that cell cycle progression into the G_2_/M phase was not inhibited. This suggests that the chromatin changes induced by chaetocin treatment did not completely block cell cycle advancement, possibly due to compensatory mechanisms or the threshold of disruption required to halt cell cycle progression.

The confinement of histones within TAs likely contributes to the generally low detectable histone levels observed in dinoflagellate nuclei [55,106], especially in “isolated” chromatins where BfC termini are severed. Accumulating evidence suggests an interconnection between telomere biology and ribosomal biology across multiple dimensions [107,108]. This includes proximity localization (open ends in dinokaryons), regulation through H3K9me3, and shared protein components. Additionally, there are notable similarities in the interactions between the NE and the cytosolic cytoskeleton via the SUN-KASH protein complex [107,109,110,111]. TEM photomicrographs (Figure 1D) and the immobility of BfCs confirm the persistence of nuclear envelope insertions (NEIs).

The increase in chromosome compaction, along with relative intrachromosomal differences [112], supports our hypothesis and suggests a general role for H3K9me3 in tissue differentiation [113], likely through different H3K9me3-marked repetitive elements. As daughter cell G1 is initiated following parent cell S-phase exit, the association of TN H3K9me3 with proliferation supports the idea that the telomeric mark, coupled with telomere length in aging, plays a role in cell proliferative conditioning across cell cycles. Myosin immunolabeling concentrated around critical domains adjacent to the NE [73], implicating telomeric nucleosomes in maintaining chromosome mobility.

Nucleosomal “occupancy” provides a snapshot of immunoprecipitable histones bound to genomic DNA, reflecting dynamic interactions, rather than a static state. The generally lower compaction and occupancy of telomeric nucleosomes [114,115] suggest a higher propensity for nucleosomal shifts due to solenoid coiling adjustments at low supercoil density. The retention of TNs likely stems from the requirement for supercoil adjustment, which, in the case of dinoflagellate chromosome ends, transmits mechanically to the TA–NE axis.

In dinoflagellates, transcriptionally active domains are primarily confined to PCLs and are distinct from the ‘Structural DNAs’ of dinochromosomes [20,40]. Restriction enzyme (RE) digestion allowed the separation of the coding fraction of the genome [39], indicating that coding regions are not protected and are unlikely to be compacted with nucleosomal regions. The increased accessibility following RE treatment suggests that inner compartments are connected to telomeric nucleosomes. Semi-decompacted nuclei exhibited enhanced cortical labeling with varying intensities, likely due to the differing degrees of decompaction within telomeric enclaves. This indicates that the compaction and decompaction of telomeric nucleosomes affect the association of chromosome ends with the NE.

The positioning of chromosomes at the NE has significant implications for the initiation and termination of replication. While the mechanisms underlying the establishment of chromosome territories (CTs) are not fully understood, the well-separated G_1_ positioning indicates that non-random positions are inherited post-telophase. Nucleosomal CTs, which delineate chromosome positioning, display a non-random and cell-type-specific organization, suggesting that differential transcription affects CT volumes [36,116,117,118]. In canonical eukaryotes, the association of telomeres with the NE regulates chromatin silencing, telomere replication, and recombination [119,120]. Telomere localization is non-random, with enrichment at the NE following mitosis [119] and in quiescent cells [120]. Variations in CTs across different cell types, which affect gene expression [121], likely exhibit feedback with karyogenomic architecture (KGA), comprising evolutionary non-neutral non-coding regions and the non-random distribution of gene-encoding domains. The formation of CTs, including those observed in BfCs, is associated with the NE matrix, suggesting that telomeric NE formation involves vesicular formation contributing to daughter nuclear envelope reformation.

Chromosomes, composed of genomic nucleic acids arranged into manageable lengths, exhibit non-random accessibility of individual domains. In this study, we employed whole-nuclei nucleosomal digestion coupled with different cross-linkers (MNRPs) to assess karyogenomic architecture (KGA). The relative Southern blot intensity of individual genes compared with standard markers offers an endogenous estimate of locus accessibility, especially for multicopy loci in sequenced genomes. Repetitive element dynamics play crucial roles not only in KGA, but also contribute to recombination potential, meiotic drive, damage propensity, and genome evolution. Understanding these complex interactions will provide deeper insights into nuclear organization and its impact on cellular processes. While nuclei undergoing NE disassembly experience substantial dynamics that are technically challenging to study in real-time [122], the dinokaryon, with its large size (>10 µm in *K. brevis*), serves as a valuable model system for investigating internuclear transport processes.

Both BfCs and TAs address genome condensation and accessibility challenges in the context of duplication and partitioning, despite dinoflagellates’ large genomes and quasi-condensed chromosomes. This offers valuable insights into chromosome engineering across species. The TN configuration provides a unique perspective on DNA packaging without conventional nucleosomes, challenging traditional views on chromatin dynamics and offering new insights into genome organization across diverse organisms. It also highlights the evolutionary adaptability of chromosome architecture and may reveal universal principles in chromosome biology, balancing DNA compaction and accessibility.

The unique TA configuration may necessitate new chromosome isolation techniques to preserve telomere associations, particularly for genome sequencing and epigenetic mapping. Additionally, the telomeric nuclear envelope insertions likely contribute to the mechanical sensitivity of many dinoflagellate species, which is relevant to their roles as symbiotic zooxanthellae in corals and as regular phytoplankton.

## 4. Materials and Methods

### 4.1. Strains and Cultures

*Karenia brevis* (strain CCMP2229, obtained from The Provasoli-Guillard National Centre for Culture of Marine Phytoplankton, Dania Beach, FL, USA) was cultured in L1 medium [123] and maintained at 18 °C under a 12 h light/12 h dark cycle. The heterotrophic dinoflagellate *Crypthecodinium cohnii* Biecheler, strain 1649, acquired from the Culture Collection of Algae at the University of Texas at Austin (USA), was maintained in MLH liquid medium [124] and cultured at 28 °C under complete darkness. Live *Karenia* cells are large, measuring 20–35 µm [10], with a flattened shape conducive to detailed observation of birefringent chromosomes (Figure 2A) and identification of size-dependent cell cycle stages.

We employed semi-synchronized cell preparations to investigate the distribution of BfCs across different stages of the cell cycle. Semi-synchronization was induced by wrapping the cultures in tin foil 48 h before sampling, then returning them to their normal light/dark cycles. Sampling started 12 h post-release, with 12 time points taken every two hours.

### 4.2. Birefringent Chromosomes

Different fixation methods can differentially alter the apparent structure and organization of nuclei chromosomal distribution, potentially leading to the misinterpretation of their native state. Kellenberger et al. (1992) demonstrated that the degree of chromosome condensation in dinoflagellates being dependent on the fixation method used [61]. Aldehyde fixation (using formaldehyde and glutaraldehyde) prior to osmium tetroxide (OsO4) fixation resulted in varying degrees of chromosome decondensation and swelling, and the apparent cholesteric chromosomal bands attributed to the low protein–DNA ratio [61,125]. We thus deploy the term birefringent chromosomes (BfCs), rather than liquid crystalline chromosomes, which is misleading in that not the whole chromosomes are liquid crystalline. Whereas freeze-substitution methods gave slapshot images, Metripol imaging and birefringence observations provide the *in vivo* native chromosome state and higher-order structures without protein tagging that affected chromosome condensation.

### 4.3. Molecular Biology and Immunological Techniques

Molecular biological techniques followed our previously published protocols [126,127,128]. Two-dimensional (2D) agarose gel electrophoresis was performed as described previously [126]. In the first dimension, DNA was separated on a 0.35% agarose gel in 1 × TAE buffer for 48 h at 0.8 V/cm at room temperature. The DNA-containing gel slice was then rotated 90° and cast into a 1% agarose gel with 0.3 μg mL^−1^ ethidium bromide in 1 × TAE.

Immunoprecipitation was performed using established protocols [129], with antibodies affinity-purified against membrane-bound antigens [130] and pre-cleaned with bacterial acetone powder [131]. Secondary antibodies were also pre-cleaned using dried dinoflagellate (*C. cohnii* or *K. brevis*) acetone extracts [131].

We employed established protocols for isolating nuclei from *Karenia* cells, which maintained birefringence for at least 5 h in isolation buffers [64]. Chromosomes could be prepared from the nuclear preparation by simple resuspension in a buffer containing a higher concentration of KCl (10 mM Tris-HCl pH 7.4, 100 mM KCl, 1.5 mM MgCl_2_, 0.1 mM DTT, 0.5% NP-40). If the nuclear envelope and nucleoplasm were still intact, the nuclear sample was then passed through a 25-gauge needle to disassociate the chromosomes from the nuclei.

Despite previous beliefs of being histoneless, recent transcriptomic–genomic analyses have identified core histones within these organisms, albeit at typically low expression levels [55,106]. To investigate this further, we developed an N-terminal-directed antibody specific to the dinoflagellate H2A histones, which have an acidic isoelectric point (pI). Based on this characteristic, we hypothesized that traditional acidic extraction buffers might be ineffective for these histones, as a basic extraction buffer (pH > pI) would make the protein negatively charged, increasing its solubility and facilitating extraction. Consistent with our hypothesis, we found that using a neutral or mildly basic buffer (pH > pI) significantly enhanced their extractability. Psoralen crosslinking with trimethylpsoralen (TMP) (Sigma-Aldrich, St. Louis, MO, USA) was conducted according to the manufacturer’s instructions. TMP, dissolved in DMSO, was added to *C. cohnii* cells at a concentration of 1 × 10^7^ cells mL^−1^ in seawater, with a final concentration of 200 µM. In our study, we applied sirtinol at concentrations ranging from 10 to 100 µM, which is 5-fold lower than that typically used for mammalian cells [132]. The treatment durations were 3, 6, 9, and 24 h, an approach designed to minimize prolonged exposure and potential off-target effects. Notably, at earlier time points (e.g., 6 h) and lower concentrations (10 and 50 µM), the effects of sirtinol were less pronounced, with no observable nuclear swelling. This observation suggests that sirtinol’s effects are threshold-dependent, rather than strictly dose-dependent. The gradual onset of observable changes indicates that a certain level of sirtinol accumulation or duration of inhibition may be necessary to induce significant alterations in nuclear structure.

Since most commercial histoneH2A antibodies adopt full proteins as immunogens, we developed a polyclonal anti-CcH2AX.2 (MN889535) antibody against the C-terminal sequence (amino acid 93–200 including the histone fold) of CcH2AX.2 that was shared with CcH2AX.1 (Appendix A). This antibody would recognize both unbound and bound CcH2As.The affinity-purified anti-CcH2AX.2 antibody, but not the pre-immune serum, recognized the recombinant CcH2AX.2 C-terminal polypeptide (Appendix A) and in *C. cohnii* cell lysate as a polypeptide band of 23 kDa (CcH2AX.1) and a band of 21 kDa (CcH2AX.2) (Appendix A).

Polyclonal anti-dHlp antibody was generated against full-length bacterial expressed CcHCc3p and should have recognized other monomeric dHLP isotypes that exhibited high homology and similar sizes (13–14 kDa) [133]. Anti-histone H2B (sc-10808), anti-histone H3 (sc-8645R), and anti-α-tubulin (sc-53646) antibodies were purchased from Santa Cruz. Anti-histone H3 (tri methyl K9) antibody (ab184677) was purchased from Abcam, Cambridge, MA, USA. All the HCc3 (dHLP), H2B, and H3 antibodies were affinity-purified with membrane-bound antigens (*Crypthecodinium cohnii* HCc3, H2B, and H3) [130] after pre-cleaning with dried bacterial acetone powder [131]. Cell lysate preparation and immunoblotting essentially followed published protocols [134,135]. Band intensities were determined using ImageJ 1.54g software (NIH, Bethesda, MD, USA) [136]. All PCR primers are listed in Appendix A.

*In vivo* crosslinking with psoralen has been widely used for nucleosome mapping [137]. Psoralen-crosslinking with trimethylpsoralen (TMP) was conducted according to the manufacturer’s instructions (Sigma-Aldrich, USA). TMP, dissolved in DMSO, was added to *C. cohnii* cells at a concentration of 1 × 10^7^ cells mL^−1^ in seawater, at a final concentration of 200 µM. Following a 30-min incubation at 22 °C under darkness, the cells were exposed to UV-A light at 365 nm using a UVGL-55 handheld UV lamp (UVP Inc., Urbana, CA, USA) from a distance of 5 cm for 10 min. This cross-linking technique was utilized to differentiate between supercoiled domains and non-supercoiled domains, as the supercoiled regions were less susceptible to nuclease activity. Our experimental conditions (200 µM psoralen, 30 min incubation) were comparable to those used in previous studies on mammalian cells (200 µM) [137] and budding yeast (2 mM, 90 min incubation) [138,139].

For double-crosslinking, the cell pellet was resuspended in 5 mL of seawater (10^7^ cells mL^−1^) and fixed with 2 mM of DSG (disuccinimidyl glutarate) for 45 min at room temperature (22 °C). Cells were double-crosslinked with 1% (*v*/*v*) formaldehyde for 30 min or UV-crosslinked with 200 µM TMP. Cross-linking was quenched by treating the samples with 125 mM glycine at room temperature for 5 min.

Unless specified otherwise, all chemicals used in the experiment were sourced from Sigma-Aldrich, USA.

### 4.4. Micrococcal Nuclease (MNase) Resistance Profiling

Micrococcal nuclease (MNase) resistance profiling is a widely established technique for analyzing nucleosomal structures due to its ability to selectively digest linker DNA while preserving nucleosome-bound DNA fragments [140,141]. The notably large nuclei of *K. brevis* (~10 µm) maintain their compacted state and exhibit birefringence in nuclei isolation buffer (NEB) [64] and allows us to perform MNase digestion under controlled conditions to assess the dosage-dependence of nuclease-resistant domains in relation to chromosomal architecture. The consistent repeatability of the MNase protection profiles (MNPP) indicated similar chromosome architecture as well as orientation relative to the NE. When comparing MNPP patterns with probes targeting different repetitive elements, and gene-coding regions (as discussed in the accompanying paper), it was found that whole-nuclei MNPPs provided specific and reproducible indications of karyogenomic localization.

*Karenia* nuclei were isolated from approximately 100 mL of log-phase culture by centrifugation at 1000× *g* for 5 min and subsequently washed in ice-cold hypotonic buffer with brief spins. The nuclear extraction buffer used contained 1 mM Tris (pH 7.4), 1.5 mM CaCl_2_, 10 mM KCl, 0.1 mM dithiothreitol (DTT), 0.5% (*v*/*v*) NP-40, and protease inhibitors including 1 mM PMSF, 20 µg mL^−1^ aprotinin, 20 µg mL^−1^ leupeptin, and 2 µg mL^−1^ pepstatin A [142,143]. For micrococcal nuclease (MNase) resistance profiling, the hypotonic buffer was exchanged for MNase buffer. The structural integrity of the nuclei in the MNase buffer was verified using polarized light microscopy, and only cells exhibiting birefringence were selected for the study. This study employed *K. brevis* cells, which are larger and can be cultivated to a higher cell concentration than *K. papilionacea* [64], enabling the visual differentiation of cell cycle phases. These distinctions are possible due to the size variation of *K. brevis* cells (approximately 20 to 35 µm) and their unique athecate, flattened cellular, and nuclear morphologies. The two species possess essentially identical karyobirefringenotypes.

Before MNase digestion, a portion of the nuclei was reserved and mixed with an equal volume of reaction termination buffer (250 mM EDTA, 100 mM EGTA), which would have led to BfC decompaction, to serve as an undigested control. MNase was added to the remaining nuclei at a concentration of 1 gel unit (approximately 0.1 Kunitz unit from NEB) per 800 nuclei. At each time point, aliquots of the digestion reaction were removed and stopped by the addition of termination buffer. Later, samples from various time points were treated with deproteination solution (0.5% SDS, 0.1 mg mL^−1^ proteinase K, 0.5 μg mL^−1^ RNAse A) and incubated at 50 °C for 2 h. Following standard phenol–chloroform–isoamyl alcohol (PCI) extraction and isopropanol precipitation with 0.3 M sodium acetate (pH 4.7), the DNA was resuspended in TE buffer at the original volume. Finally, both native and deproteinated MNase-digested *K. brevis* nuclei were analyzed by electrophoresis on 1.5% agarose gel to evaluate the MNase resistance profile. Considering the biological variations among different nuclear preparations, samples were additionally probed with repetitive elements cc18 and cc20 [74], which have been shown to exhibit distinct AFLP patterns [75], with cc18 specifically localized within the surface-associated gene-coding compartment.

### 4.5. Polarized Light Microscopy with Semi-Automatic Installation

Birefringence microscopy of live *Karenia brevis* cells [10], which have a flattened shape conducive to detailed observation, was conducted (Figure 2A). The PLM setup and protocols adhered to those described previously [10]. The details of the birefringent microscopy are described in the accompanying genome duplication–partitioning paper. The slides were examined under crossed nicols using an Olympus BX51 microscope, which was equipped with a Brace–Kohler (U-CBR2) compensator set to +20° lambda to enhance contrast. Photomicrographs were captured with a Pixera Penguin 600 CL digital camera attached to the microscope. The imaging utilized a 40× UplanFI objective and a 100x ACH objective, paired with a 0.9 NA Achromat condenser (U-POC-2). For each cell cycle time point, approximately ten photomicrographs were taken at 40× magnification to capture as many cells as possible. These images were then manually screened to exclude any cells showing signs of physical damage before measurements. We demonstrated the Metripol system’s ability to detect changes in birefringence due to DNA decondensation induced by EDTA in *K. brevis* [64]. Our previous publication provides a comprehensive discussion on using Metripol to measure birefringence across different dinoflagellate species [10]. To prepare slides for time-lapse photomicrography, a series of steps were followed to avoid sample drying and to restrict the motion of the cells. Slides were cleaned with ethanol before a thin bead of Halocarbon oil 700 (Sigma-Aldrich, USA, H8898) was applied to form a square barrier on the slide’s center. A 5–10 µL sample was pipetted from the culture flask onto the oil, and a coverslip was gently placed on top. Images were taken at one-minute intervals as long as the subject remained stationary within the field of view. Throughout the time-lapse photomicrography, the light intensity was carefully controlled at a minimum level. Subsequently, individual frames were exported and compiled into a cohesive movie format, which was then analyzed for chromosomal movement.

We adopted the term birefringence chromosomes (BfCs) to highlight a different liquid crystalline model from the previously proposed higher-order cholesteric model, which was attributed to artefactual decompaction due to inefficient protein–nucleic acid fixation, leading to the partial decompaction of the plectonemic structure [61]. This term is also misleading because not the whole chromosome is liquid crystalline, which would have implications for chromosome dynamics. Our psoralen cross-linking and resulting periodicity supported the existence of plectonemic modules, which we termed supercoil modules (SPMs). These SPMs were aligned anisotropically and superimposed on the inner compartments. The birefringent nature of these structures would be evident when examining individual layers under circularly polarized light. As G_2_ BfCs underwent lengthening with surface modifications, the artefactual effect of fixation would differentially alter the G_1_ BfCs more than those in S-G_2_. Consequently, the sectional–segmental demarcation observed in fixed samples would reflect the endogenous arrangement, despite some decompaction of the inner plectonemic modules.

### 4.6. Transmission Electron Microscopy and Sample Preparations

*Karenia* cells were prepared for ultrastructural analysis using a standard protocol of glutaraldehyde–osmium tetroxide fixation. The culture was pretreated at 1% glutaraldehyde and pre-cooled on ice before collection by centrifugation at 3000× *g* for 10 min. The cell pellet was then carefully resuspended in 4% glutaraldehyde with 0.1 M PIPES buffer and left for at least 1 h at 4 °C to preserve the overall chromosome structures prior to secondary fixation. This procedure was designed to maintain the cellular structure without the damage often caused by centrifugation-induced ecdysis. Following initial fixation, the cells were rinsed twice in 0.1 M PIPES buffer for 15 min each to remove excess glutaraldehyde. A secondary fixation was performed with 1% osmium tetroxide in 0.1 M PIPES buffer for 1 h at 4 °C in the dark.

Subsequently, the cells were dehydrated through a graded ethanol series, from 10% to 100%. They were then embedded in Spurr’s epoxy resin (Sigma-Aldrich, USA) and sectioned into ultra-thin slices [67,142,144]. For contrast enhancement, the sections were stained with 1% uranyl acetate in 50% methanol and 0.25% lead citrate in 0.1 N NaOH for 15 min each. The stained sections were mounted on carbon-coated copper grids for examination with a JEOL 100 CX transmission electron microscope. The sample fixation and cross-linking agents resulted in varying degrees of chromosome decompaction, which was exploited for investigating chromosome ends (Figure 1D).

## Figures and Tables

**Figure 1 ijms-25-11312-f001:**
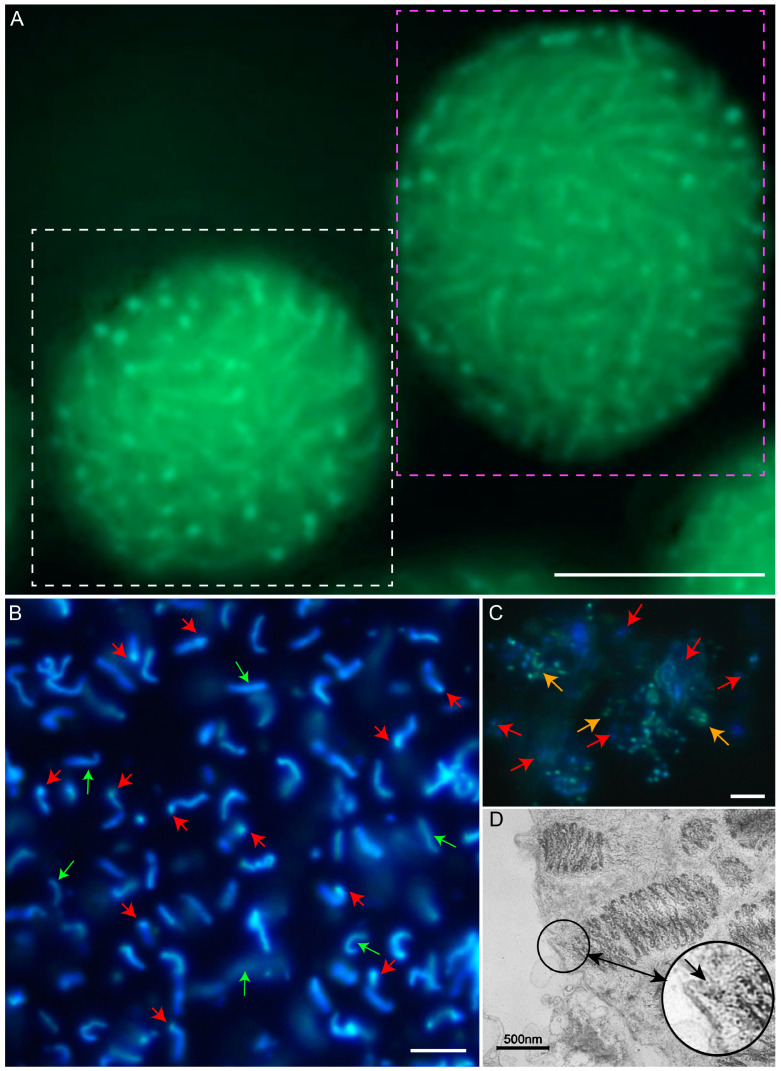
Fluorescent photomicrographs of isolated *Karenia* nuclei and chromosomes exhibiting differential decompaction between the two termini. (**A**) Fluorescent photomicrograph of freshly prepared *Karenia* nuclei examined with SYTOX Green, revealing well-separated chromosomes with unstainable peripheral chromosome loops. Certain dyes were less effective at staining the chromosomes, possibly because their intercalation led to increased decompaction. G_1_ cell—white dashed rectangle, G_2_ cell—pink dashed rectangle. (**B**) Chromosome preparations stained with DAPI showed notable differences at the chromosome ends, with the condensed ends marked by red arrows. Several isolated chromosomes appeared to lack a condensed domain at either end (green arrows). (**C**) After digestion with micrococcal nuclease, most BfCs were digested, leaving behind residual dots (indicated by red arrows). A smaller BfC fraction did not stain well with DAPI, exhibiting a green color; these chromosomes were longer and occasionally had sister BfCs joined together (orange arrows), considered to be G_2_ BfCs that required duplication prior to segregation. Some dots appeared green, unlike the typical blue of DAPI-stained DNA. We interpret these green signals as membrane lipids associated with telomeric anchorage to the nuclear envelope. This interpretation is based on several factors: their resistance to nuclease digestion, which suggests a non-DNA composition; the shift from blue to green fluorescence, which is consistent with DAPI’s known interaction with lipids [65]; and their localization, which aligns with the expected position of telomere attachment sites on the nuclear envelope. While further investigation using specific lipid stains and telomere probes would be beneficial to confirm this interpretation, the observed characteristics are consistent with lipid-rich structures. The chromosomal material that remained resistant (indicated by orange arrow) is thought to represent mitotic anaphase BfCs with a modified chromosome surface. The less-stained chromosome ends, with the proposed anaphase resistance domain, suggested either reduced accessibility or compaction of the mitotic telomeric regions. (**D**) Transmission electron microscopy of a *Karenia* nuclear section revealed chromosome ends (black arrow) on the nuclear envelope. Not all DNA dyes, which differed in their effects on DNA structures, resulted in staining enclaves on the NE. For (**A**–**C**), scale bar = 10 μm.

**Figure 2 ijms-25-11312-f002:**
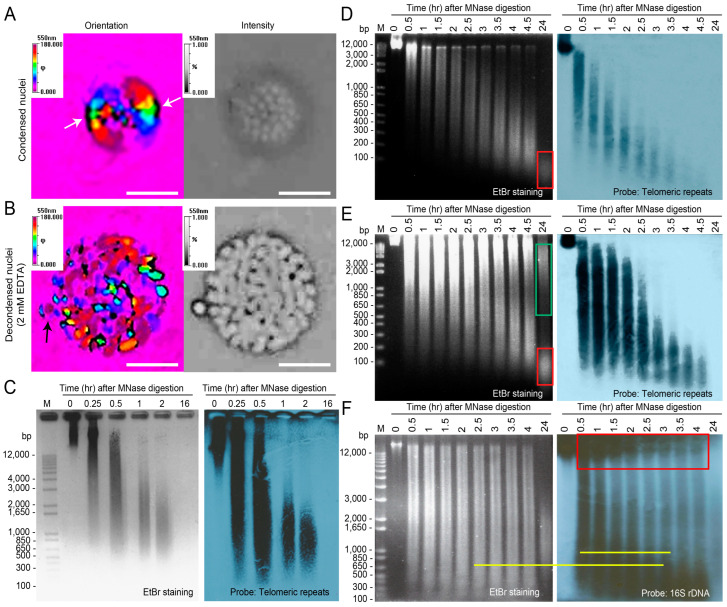
Comparative micrococcal nuclease profiling suggesting that two chromosome termini have rDNA and telomeric repeats with different resistance. (**A**,**B**) Semi-automatic polarized light imaging (Metripol) of isolated nuclei suggesting non-nucleic acid domains (black, indicated by white arrow) at chromosome termini in the nuclear envelope that persisted after cation chelation. Chromosome decompaction was induced by treating the nuclei with (**B**) 2 mM EDTA. The orientation micrographs depict extinction angles, which correspond to the angle between the long axis of the chromosomes and the alignment direction of the chromatin filament fibers. Black arrow points to one chromosome end expulsed from the NE. Scale bar = 10 μm. (**C**) Ethidium bromide staining and telomeric Southern blot analysis of *K. brevis* nuclei subjected to MNase digestion following restriction enzyme treatment (25 U MboI digestion of 10^5^ nuclei). (**D**) MNase-digested isolated chromosome preparation, and (**E**) MNase-digested nuclei preparation. The isolated chromosomes did not exhibit the prominent MNase-protected domains that the whole nuclei did (red boxes, Figure 2D compared to Figure 2E), with apparent m.w. lower than 100 bp, and the protected domain remained largely undigested, even after 24 h (compared to digested nuclei in Figure 2E, green box region). (**F**) Ethidium bromide staining and 16S rDNA Southern blot analysis of MNase-digested *K. brevis* nuclei. The high-molecular-weight rDNA in the open-end (red box) did not share the resistance pattern of telomeric repeats (**E**), indicating that some of the open-ended rDNAs were in a nucleosomal conformation that persisted for up to 5 h of MNase digestion. MNase-protected domains, revealed by overnight digestion (24 h) and probed with rDNA, indicated the presence of persistently protected rDNA loci, implicating potential protection with nucleosomal domains. Two distinct rDNA (rRNA)-associated populations are highlighted in yellow (**F**), with equal intensities, suggesting that the population associated with the gel was not a result of digesting the well-associated population.

**Figure 3 ijms-25-11312-f003:**
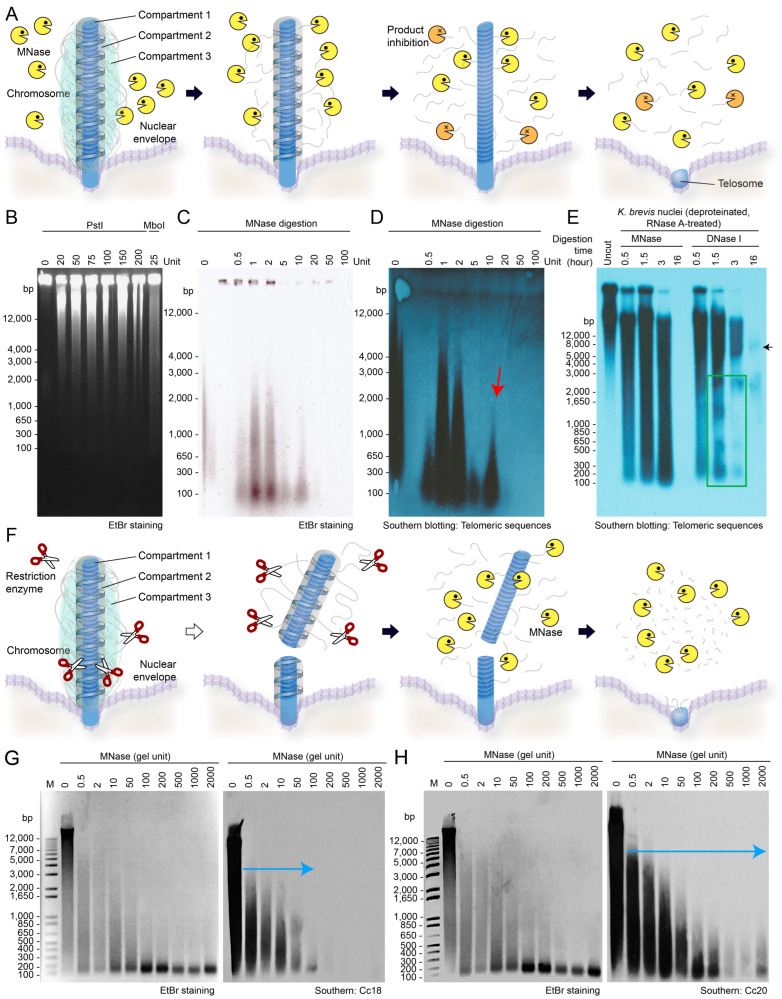
High-resolution analysis of micrococcal nuclease-resistant regions revealing DNA–protein interactions and chromosomal anchoring structures. (**A**) Diagrammatic representation of micrococcal nuclease-mediated BfC disassembly in the absence of restriction enzymes. BfCs of dinoflagellates are hypothesized to organize into three distinct compartments [63]: Compartment 1 (C(i)) is the highly condensed inner core with a columnar–hexagonal mesophase, exhibiting the highest DNA packaging density. Compartment 2 (C(ii.1)) is the less dense surface layer that spirals around C(i), consisting of chromonema coils. Compartment 3 (C(ii.2)) forms the outermost layer, comprising peripheral chromosomal loops (PCLs) that are transcriptionally active. C(ii.1) and C(ii.2) exist in a dynamic equilibrium, with C(ii.2) capable of condensing into C(ii.1), especially during mitosis. This compartmental organization allows for unique structural flexibility and functional regulation through soft-matter phase transitions, which are crucial for the distinctive chromosome architecture and function in dinoflagellates [63]. (**B–D**) Pre-digestion of additional restriction enzymes led to faster emergence of the higher-sensitivity fraction and more resolved higher resilient fractions. (**B**) Isolated *Karenia* nuclei were digested with PstI (CTGCA^G) and MboI (^GATC) at 37 °C for 1 h prior to gel electrophoresis. (**C**) *Karenia brevis* nuclei digested with MboI were subsequently subjected to MNase digestion for 1 h using various amounts of the enzyme (enzyme units). (**D**) Southern blotting of the samples from (**C**), using telomeric sequences as the probe. Telomeric positive resilient fractions were observed at lower MNase amounts (0.5–10 units). We interpret this as potentially indicative of mitotic chromosomes with highly condensed domains. The increasing amount of telomeric Southern signal, rather than decreasing as expected for open accessibility, suggested that the NE-associated chromosome ends were only accessible after concerted RE + micrococcal nuclease digestion (red arrow). (**E**) Deproteination of the DNase I-digested sample resulted in a nucleosome-like ladder pattern (green box) on the telomeric Southern blot. (**F**) Diagrammatic representation of micrococcal nuclease-mediated BfC disassembly with restriction enzymes. Southern blotting analysis of *C. cohnii* nuclei MNase profiling using probes specific to (**G**) repetitive elements Cc18 and (**H**) Cc20.

**Figure 4 ijms-25-11312-f004:**
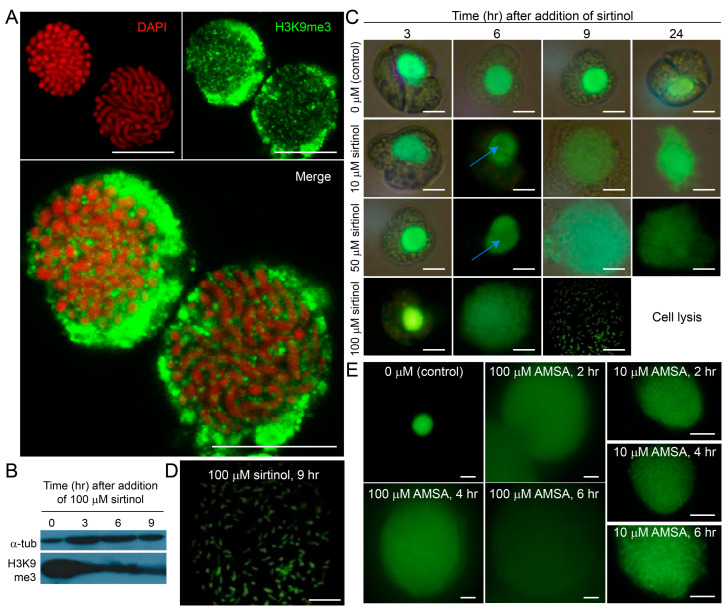
Sirtinol-mediated chromosome decompaction leading to telosomal enclave expansion within nuclear envelope. Sirtinol treatment resulted in the decompaction of chromosome termini, accompanied by the formation of swollen-like nodules on the nuclear envelope. (**A**) Confocal images of anti-H3K9me3-labeled *Karenia brevis* nuclei. DAPI-stained DNA was pseudo-colored red. Although no apparent labeling was observed at the nuclear center, the nuclear envelope (NE) may have been displaced during post-fixation, resulting in the appearance of a thicker NE. (**B**) Immunoblot analysis of the H3K9me3 epigenetic mark in *Karenia brevis* cells treated with 100 μM sirtinol. (**C**) Sirtinol-induced decompaction of chromosome termini. After 9 h of treatment with 100 µM sirtinol, the nuclear volume significantly increased, ballooning to the size of the cells. (**D**) At 100 µm sirtinol, individual BfCs, initially tightly packed, became visible. By T = 9 h, BfCs lost their stainability, except at the nuclear envelope, where swollen-like nodules were observed. All the samples were stained with SYTOX Green. (**E**) Comparison of control and AMSA-treated isolated nuclei: Control, 2 h AMSA treatment, 4 h AMSA treatment, and 6 h AMSA treatment. AMSA-induced BfCs exhibited much greater decompaction, without individualized BfCs being visualized, and no SYTOX Green staining was observed on the nuclear envelope. This suggests that the decompaction of BfCs mediated by the topoisomerase II inhibitor AMSA progressed from the chromosome proper, whereas sirtinol-induced decompaction progressed from the nuclear envelope. It is noteworthy that sirtinol required a significantly longer time for decompaction compared with AMSA, with centrally located BfCs taking a longer time to decompact (blue arrows). Scale bar = 10 μm.

**Figure 5 ijms-25-11312-f005:**
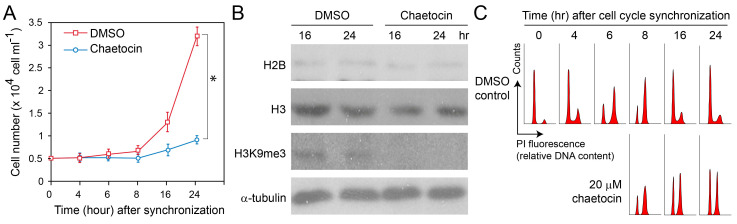
H3K9me3 reduction suppressed the exit of the S phase and cell proliferation. (**A**) Chaetocin treatment was associated with reduced cell proliferation. Data represent means ± SE of triplicate experiments. Asterisks (*) indicate significant differences from the control (*p* < 0.05). (**B**) Immunoblot analyses of core histones in chaetocin-treated *C. cohnii* cells confirming the inhibition of H3K9me3. (**C**) DNA-flow cytograms of synchronized *Crypthecodinium cohnii* cells treated with chaetocin (administered at T = 0). Chaetocin, a histone methyltransferase inhibitor specifically targeting histone H3K9 tri-methylation (H3K9me3), induced a delay in S-phase exit (or G_2_ phase entry) in treated cells compared with the vehicle control (0.01% *v*/*v* DMSO), evidenced by the gradual shift of twin peaks toward the G_1_ phase.

**Figure 6 ijms-25-11312-f006:**
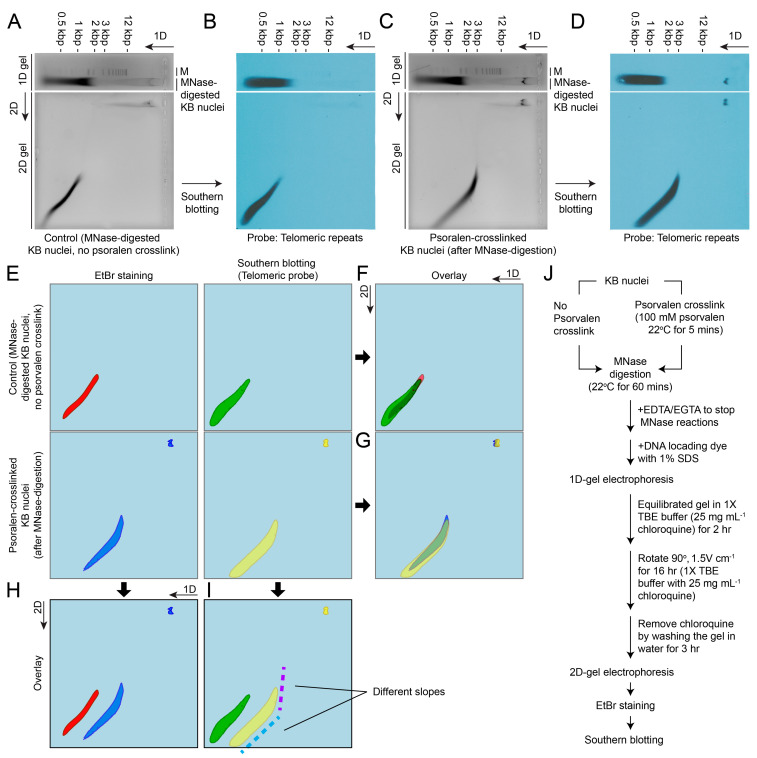
Telomeric nucleosomal domains are the major core histone-containing fraction in birefringent chromosomes. (**A**) Ethidium bromide (EtBr) staining and (**B**) telomeric Southern blot analysis were conducted on 2D gels analyzing the MNase-digested *K. brevis* nuclei and (**C**,**D**) psoralen-crosslinked, MNase-digested *K. brevis* nuclei. Psoralen crosslinking led to a complete shift in resistance of all telomeric sequences to MNase digestion. (**E**–**I**) Overlay images of the EtBr-stained gels and telomeric Southern blots. Psoralen pre-crosslinking caused a significant shift in nearly all resistant DNAs, including two populations of telomeric-containing DNA. The higher-molecular-weight fraction, which was stained with EtBr, indicated the presence of dsDNA. The EtBr-negative fraction may consist of ssDNA at lower concentrations. Both populations shifted to the same position following psoralen treatment, suggesting their association with proteins. The majority of the telomeric repeat-positive range exhibited a linear relationship, indicating a dose-dependent effect. Most of the supercoiled DNA exhibited resistance to MNase digestion and contained telomeric sequences. The substantial changes in mobility demonstrated the lesser superhelicity commonly associated with telomeric nucleosomes. (**J**) Workflow illustrating the procedure for MNase digestion and psoralen crosslinking of *K. brevis* nuclei.

**Figure 7 ijms-25-11312-f007:**
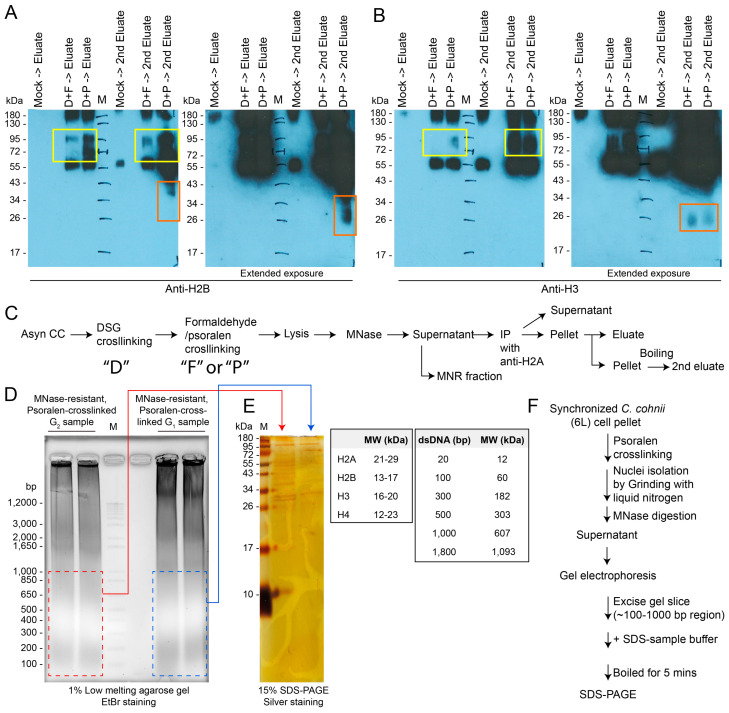
Psoralen pre-crosslinking led to a substantial increase in immunocaptured micrococcal nuclease-resistant nucleosome–octameric complexes. (**A**) Immunoblot analysis of histone H2B in the CcH2A-immunoprecipitates pulled down from DSG–formaldehyde or DSG–psoralen-crosslinked *Crypthecodinium cohnii* cell lysates. (**B**) Immunoblot analysis of histone H3 in the same immunoprecipitants as (**A**). Samples were analyzed on denaturing SDS-PAGE gels. The use of psoralen coupled with DSG-crosslinking, when compared with DSG–formaldehyde crosslinking alone, was more efficient in pulling down high-molecular-weight bands/complexes (yellow boxes) that were absent in the mock control. These high-molecular-weight bands represent crosslinked protein complexes that were not fully dissociated under the denaturing conditions. Lower-molecular-weight bands/complexes (orange boxes), likely corresponding to H2A–H2B or H3–H4 complexes, were also detected in the IP product. (**C**) Workflow diagram for (**A**,**B**). (**D**) EtBr staining of electrophoresed Psoralen-crosslinked G_1_ and G_2_ *C. cohnii* nuclei preparations. (**E**) Silver staining of the excised ~100–1000 bp fraction (from (**D**)) following SDS-PAGE. Notably, the apparent molecular weight (mw) of several kilobase pairs shifted from the monomeric 100–200 bp DNA (without proteins) to approximately 500 bp of protein–DNA complex within a nucleosome. The free population with open ends exhibited a streaky and continuous distribution, while the non-open end displayed higher integrity. (**F**) Workflow diagram for (**D**,**E**).

## Data Availability

Data are contained within the article.

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
