# Peer review of "Dinochromosome Heterotermini with Telosomal Anchorages"

_ijms, 2024, doi:10.3390/ijms252011312_

Round 1

Reviewer 1 Report

Comments and Suggestions for Authors

Dear Authors,

This study presents a significant advancement in understanding chromosome function, particularly in the context of dinoflagellates, organisms known for housing some of the largest genomes. The paper emphasizes how chromosome positioning affects key functions such as replication, segregation, differential gene expression, and DNA damage susceptibility. The focus on the structure and behavior of birefringent chromosomes (BFCs) in dinoflagellates, including their interaction with the nuclear envelope (NE) through telomeric nucleosomes, offers a novel and valuable contribution to genomic research. This aspect is particularly critical when chromosome ends are situated in telomeric regions or nuclear territories. The study also highlights the role of histone H3K9me3 in chromosome compaction and telomeric nucleosome-mediated stabilization of chromosome ends.

The research on Karenia brevis and other dinoflagellates with BFCs will add further depth by discussing how these modifications influence cell division and telomere length maintenance. Overall, this work will contribute to advancing academic knowledge in organisms with unique chromosomes, particularly in algae species.

Revision Points:

1.      The overall length of the paper, particularly in the Results section, and the complexity of its explanations, makes it difficult for readers to extract key information. Sections marked in blue (as indicated in the PDF) are notably lengthy, making it challenging to distinguish between the explanation and the Discussion. Considering the relatively small number of researchers in this specific field, I recommend streamlining the content to improve the flow and accessibility of the results. For instance, the sections marked in blue could benefit from simplification. Clearer articulation of each section's objectives, the reasoning behind the experiments, and the presentation of results will enhance readability and the overall structure of the paper.

2.      While the paper covers various techniques like Comparative Micrococcal Nuclease profiling and the use of chemical agents such as Sirtinol (a promoter histone H3 acetylation and decompaction chromosomal termini), Chaetocin (a histone methyltransferase inhibitor), and Psoralen (a cross-linkeï½’ supercoiled DNA), it would help to more explicitly explain why these methods were chosen.  The use of chemical agents such as Sirtinol, Chaetocin, and Psoralen may raise concerns about overall toxicity. It might be useful to consider ways to limit their action to specific functions, potentially addressing this issue in the manuscript.

3 .     While the study presents novel insights into dinoflagellate chromosomes, expanding on the broader implications of the findings would further enhance its impact. How do these findings contribute to our universal understanding of chromosomal biology in other organisms? What future research directions could stem from these results? Addressing these questions would significantly strengthen the paper's contribution to the field.

I have added further comments to the PDF for your reference, which should assist in the revisions necessary to make this work suitable for publication in IJMS.

Author Response

Point-by-point response to Comments and Suggestions for Authors

Comment 1: The overall length of the paper, particularly in the Results section, and the complexity of its explanations, makes it difficult for readers to extract key information. Sections marked in blue (as indicated in the PDF) are notably lengthy, making it challenging to distinguish between the explanation and the Discussion. Considering the relatively small number of researchers in this specific field, I recommend streamlining the content to improve the flow and accessibility of the results. For instance, the sections marked in blue could benefit from simplification. Clearer articulation of each section's objectives, the reasoning behind the experiments, and the presentation of results will enhance readability and the overall structure of the paper.

Response 1:

In response to the comments, we have made the following revisions:

  • To address concerns about the paper's length and complexity, particularly in the Results section, we have carefully rewritten the relevant sections (especially those highlighted) by using more concise language and reorganized the content in a more logical order to condense and simplify the explanations
  • We have shortened, reorganized and rewritten the whole Discussion section to improve its flow and clarity
  • We have added some explanatory notes to several figure legends (e.g., Figures 3A, 6, and S4) to clarify specific terms and key points
  • We have revised some Figure title, to better coney the significance of the Figure

We hope these changes would significantly improve the accessibility of our work.

The section/paragraph/statement marked in blue (shown below) by Reviewer 1 have been revised as follows:

Line 132-136: “DNA dye staining, which only visualize condensed DNA, showed that the brightly fluorescent structures, despite their coalescence, represent the inner compartment of the chromosome that did not fully decondense following chelation or during processing [1]. The peripheral chromosome loops (PCLs) [2], which did not exhibit fluorescence from the DNA dye, appeared as birefringent voids that display 'apparent gaps' between chromosomes.”

Response:

The revised version is as follows:

“DNA dyes like DAPI, which primarily visualize condensed DNA, revealed bright fluorescent structures representing inner chromosome compartments that remained partially condensed after chelation or processing. In contrast, peripheral chromosome loops (PCLs) [2] appeared as non-fluorescent, birefringent voids, creating 'apparent gaps' between chromosomes.”

Line 190-202: “Isolated BFCs displayed a distinctive stage-2 partial decompaction that was induced by cation chelation [1], with an elongated, screw-like structure wrapped around a central axis. This suggested a consistent pattern of anchorage-dependent spiraling along the BFC axis. In contrast, a motif without anchorage would display a more erratic pattern of decompaction. The G2 chromosomes exhibited more complex behavior, which will be reported in a separate study.

DNA dyes, which only visualize condensed DNA, revealed brightly fluorescent structures, despite coalescence, represent the inner compartment of the chromosome that did not fully decondense following chelation or during processing [1]. In contrast, the peripheral chromosome loops (PCLs) [2], which did not exhibit fluorescence from the DNA dye, appeared as birefringent voids that display 'apparent gaps' between chromosomes. This contrasts with the elongated shape of the isolated chromosomes, suggesting that decondensation occurs at chromosome ends rather than nucleolar ends.”

Response:

The paragraphs (in section 2.2) were deleted.

Line 227-229: “Dinoflagellate nucleoli did not disassembly during the cell cycle and were formed with nucleolar organizing center (NOC) of the corresponding chromosomes [3, 4].”

Response:

The sentence was deleted.

Line 245-259: “Previous studies have reported that restriction enzymes preferentially release gene-encoding domains, suggesting that the Perichromosomal Loop (PCL) outer compartment [5] is more readily separated from the more resilient inner compartments [6]. In the current study, the use of restriction enzymes resulted in a more rapid elimination of the middle segments, indicating that a potential resilience domain was released, correspondingly allowing quicker access to the telomeric anchorage. This aligns with previous findings that restriction enzymes increase the accessibility of chromosome ends, as demonstrated by prior TRAP assays [7]. These suggested a connection between the telomeric regions and the inner compartments. Our prior study indicated that Bal31 nuclease digestion released telomeric sequences within the relatively long size range of 50-70 kbp [14], suggesting the existence of similarly sized telomeric segments likely originating from unprotected chromosome ends. This large size range was unexpected and hinted at the relative inaccessibility of the NE to the nuclease during the normal digestion process. Assuming NE insertion, the cutting by restriction enzymes would expose the inner compartments to MNase, theoretically resulting in a faster MNRP.“

Response:

The revised version is as follows:

“Restriction enzymes preferentially release gene-encoding domains, suggesting easier separation of the PCL outer compartment [5] from resilient inner compartments [6]. Our study showed quicker elimination of middle segments by restriction enzymes, indicating release of a potential resilience domain and faster access to telomeric anchorage. This aligns with previous TRAP assays showing increased accessibility of chromosome ends [7], suggesting a connection between telomeric regions and inner compartments. Prior Bal31 nuclease digestion released unexpectedly large (50-70 kbp) telomeric sequences [7], implying inaccessibility of the nuclear envelope (NE) during normal digestion. Assuming NE insertion, restriction enzyme cutting would expose inner compartments to MNase, theoretically resulting in a faster MNRP.”

Line 316-319: “The persistence of dinoflagellate nucleoli in the cell cycle was demonstrated in the heterotrophic dinoflagellate Crypthecodinium cohnii, with nucleoli formation attributed to Nucleoli Organizing Centre (NOR)-rDNA loci situated at chromosome ends [3, 4].”

Response:

We have rewritten the whole the relevant paragraph in section 2.4, and the revised version is as follows:

“Dinoflagellate nucleoli persist throughout the cell cycle, as demonstrated in Crypthecodinium cohnii, due to Nucleolar Organizing Center (NOC)-rDNA loci at chromosome ends [3, 4]. This persistence, along with B-end chromosome ends associating with nucleoli, suggests a nucleoli-NE axis in chromosomes. Interestingly, rDNA showed a different resistance pattern than telomeric repeats (Figure 2E, F), implying some open-end rDNAs were in a protected conformation resistant to MNase digestion for up to 5 hours. The gradual decrease of the high m.w. fraction (Figure 2F, red box) over 24 hours indicated a substantial number of chromosome ends being associated with rDNAs.”

Line 338-343: “In addition to the high m.w. and low m.w. ribosomal DNA fractions, we observed streaks that reacted to the rDNA probes, indicating lower resilience binding. This points to the possibility that other nucleolar macromolecular complexes [8], potentially including ribosomal nuclear particles (RNPs) as described by Walker (1980) [9], and small nuclear RNPs (snRNPs) observed in dinoflagellates in other studies [10], contribute to the protected conformation.”

Response:

The revised version is as follows:

“Besides high and low m.w. rDNA fractions, we observed streaks reacting to rDNA probes, suggesting lower resilience binding. This indicates that other nucleolar macromolecular complexes [8], such as ribosomal nuclear particles (RNPs) [9] and small nuclear RNPs (snRNPs) [10], may contribute to the protected conformation in dinoflagellates.”

Line 453-458: “The C. cohnii H3 variants do not share the "AAIG" sequence conserved in human H3.3, nor do they possess the "SAVM" motif characteristic of the canonical H3 in humans. Instead, these variants display a diverse "Q/E/S-A/G-I/L-I/S/L/E" motif. Moreover, most (5 out of 6) of these C. cohnii variants exhibit similar transcription levels in G1 and S-G2 cells (data from in-house cell cycle transcriptome), as in the unchanged human H3.3 expression pattern [11-13].”

Response:

Because of the additional H3 information provided in Figure S4, the paragraph is revised as follows:

“Moreover, C. cohnii H3 variants do not share the "(A/Q/E/T/H)A(I/L/V)(L/G)" sequence motif conserved in many eukaryotic H3.3 proteins, nor do they possess the "SAV(M/L/A)" motif characteristic of canonical H3 in many organisms. Instead, these variants display a diverse "(Q/E/S)(A/G)(I/L)(L/S/E)" motif, which more closely resembles the H3.3 "(A/Q/E/T/H)A(I/L/V)(L/G)" motif (Figure S4). Despite these sequence differences, both CcH2A and CcH3 variants were still recognizable by commercially available H3 and H2A antibodies used in this study (Figures S1, S2, and Figure 6), which target the highly conserved histone fold region shared by canonical histones and histone variants. Furthermore, most (5 out of 6) of these C. cohnii variants exhibit similar transcription levels in G1 and S-G2 cells (data from in-house cell cycle transcriptome), reminiscent of the expression pattern of H3.3 in other eukaryotes [11-13].”

Line 467-473: “Sirtinol, an inhibitor of sirtuin deacetylases such as SIRT1, indirectly promotes the acetylation of histone H3 at lysine 9 (H3K9ac), a modification that is associated with active transcription and a more open chromatin state [14]. Although sirtinol doesn't directly alter histone methylation, including trimethylation at the same lysine residue (H3K9me3), sirtinol's enhancement of acetylation can reduce methylation. The balance between acetylation and methylation on H3K9 is crucial for the regulation of gene expression and can be influenced by sirtinol through its modulation of sirtuin activity [15].”

Response:

The revised version is as follows:

"The interplay between histone H3 lysine 9 trimethylation (H3K9me3) and acetylation (H3K9Ac) represents a critical epigenetic switch that governs chromatin state and gene expression [16, 17]. These two modifications are mutually exclusive on the same lysine residue and exhibit a reciprocal inhibitory relationship, where each modification can suppress the establishment of the other [18, 19], often associated with opposing chromatin states [20]. In mammalian cells, H3K9Me3 is typically linked to heterochromatin formation and gene silencing, while H3K9Ac is associated with (open and accessible) euchromatin and active gene transcription [21]. Sirtinol, a sirtuin deacetylase inhibitor [15], was employed to investigate the role of histone deacetylation in dinoflagellates chromatin compaction and NE integrity. In various eukaryotic systems, including plant and mammalian cells, sirtinol indirectly promotes H3K9 acetylation, a modification associated with active transcription and open chromatin states [14, 22]. While not directly altering histone methylation, sirtinol's enhancement of acetylation can reduce methylation at the same residue. "

Line 508-511: “While H3K9 acetylation (H3K9acetyl) is typically associated with transcriptional activation and H3K9me3 with repression, we investigated the distinct yet interconnected roles of H3K9acetyl and H3K9 tri-methylation (H3K9me3) in both synchronous and asynchronous cell populations.”

Response:

The revised version is as follows:

“We examined the roles of H3K9acetyl and H3K9me3 in both synchronous and asynchronous cell populations, exploring their distinct yet interconnected functions.”

Response to the specific comment raised in the Reviewer 1 pdf file:

Line 110-115: “Comparing the extraction of nuclear proteins, immunoblot analysis suggested that most of the histone CcH2B was present in the cell pellet, while dHlp (and α-tubulin) were predominantly found in the supernatant (Figure S1E). This indicated that most core histones were, surprisingly, localized in a different compartment, with minimal presence in the free chromosome-associated dHlp compartment. This observation prompted us to investigate the compartmental location of dinoflagellate core histones.“

This expression is abrupt, and it would be better to explain it in procedure that follows the flow of the experiment’s results.

Response:

Thanks for pointing this out. To improve this, we have restructured the paragraph to provide a clearer progression of ideas, following the experimental flow and gradually building up to the conclusion and subsequent research question.

Here is the revised version:

“Immunoblot analysis of extracted nuclear proteins revealed an unexpected distribution: Crypthecodinium cohnii histone H2B (CcH2B) was predominantly in the cell pellet, while dinoflagellate histone-like proteins (dHlp) and α-tubulin were mainly in the supernatant (Figure S1E). This distinct distribution suggests that core histones and dHlp occupy separate compartments, unlike their typical co-localization on chromosomes in other eukaryotes.”

Line 111: “…. most of the histone CcH2B was present in the cell pellet …..”

What is Cc?

Response:

Cc” stands for Crypthecodinium cohnii, and the sentence has been revised:

“….Crypthecodinium cohnii histone H2B (CcH2B) ….”

Line 111: “…. while dHlp (and α-tubulin) were predominantly found in the supernatant …..”

Write full name (for dHlp).

Response:  

The sentence has been revised:

“…. while dinoflagellate histone-like proteins (dHlp) and α-tubulin were ….”

Line 116-123: “Comparing the extraction of nuclear proteins, immunoblot analysis suggested that most of the histone CcH2B was present in the cell pellet, while dHlp (and α-tubulin) were predominantly found in the supernatant (Figure S1E). This indicated that most core histones were, surprisingly, localized in a different compartment, with minimal presence in the free chromosome-associated dHlp compartment. This observation prompted us to investigate the compartmental location of dinoflagellate core histones.

Time-lapse birefringence microscopy revealed no significant positional changes in the G1 chromosome (Video S1), with one end of the chromosome embedded within the nuclear envelope (NE) and the other end being “open” and extending into the nucleolar region. This affirmed our previous findings and the hypothesis that BFCs consist of inner and outer compartments [23, 24]. Continued recordings also showed that the nucleoli remained apparently unchanged throughout the observation period. This stable chromosomal configuration and nucleoli has profound implications for chromosome dynamics and the isolation of chromosomal nucleic acids for analytical purposes.”

The connection between these two paragraphs and the results is difficult to understand. In the first paragraph of the results section, it would be clearer to organize and explain what hypothesis the experiment aimed to prove.

Response:

We agree with the reviewer’s comment. To address these issues, we have restructured the paragraph as follows:

“Immunoblot analysis of extracted nuclear proteins revealed an unexpected distribution: Crypthecodinium cohnii histone H2B (CcH2B) was predominantly in the cell pellet, while dinoflagellate histone-like proteins (dHlp) and α-tubulin were mainly in the supernatant (Figure S1E). This distinct distribution suggests that core histones and dHlp occupy separate compartments, unlike their typical co-localization on chromosomes in other eukaryotes.

Time-lapse birefringence microscopy revealed stable G1 chromosome configurations, with one end anchored to the nuclear envelope and the other extending into the nucleolar region (Video S1), supporting our previous findings of inner and outer compartments within BFCs [23, 24]. Together, these results indicate that dinoflagellates possess a unique nuclear organization in which core histones and dHlp reside in separate compartments. This arrangement is likely linked to the stable chromosome structure observed throughout the cell cycle and has significant implications for chromosome dynamics and nucleic acid isolation methods in these organisms.”

Line 155-158 (Figure 1_legend): “….  (C) After digestion with micrococcal nuclease, most BFCs were digested, leaving behind residual dots (indicated by red arrows). Some dots appeared green, unlike the typical blue of DAPI-stained DNA, which we interpret as being surrounded by membrane lipids.  …..”

What do these green signals represent?

Response:  

We appreciate the reviewer's question regarding the green signals observed after micrococcal nuclease digestion. We believe these green signals likely represent membrane lipids associated with telomeric anchorage points on the nuclear envelope. This interpretation is supported by the following:

  1. DAPI staining characteristics: While DAPI typically fluoresces blue when bound to DNA, it is known to produce yellow to green fluorescence when interacting with lipids [25, 26].
  2. Nuclease resistance: The persistence of these structures after micrococcal nuclease treatment indicates they are not primarily composed of DNA, consistent with a lipid-based structure.
  3. Nuclear envelope association: In dinoflagellates, chromosomes are known to attach to the nuclear envelope at their telomeres (likely involves membrane-associated proteins and lipids). This has been demonstrated in various studies [3, 27, 28], as well as in Figure 1D.
  1. Lipid-specific PATMAN staining of MNase-resistant nuclear samples (Figure S2C) revealed concentrated dots surrounded by blue-fluorescent membrane lipids. This pattern aligns with our observations of green residual dots after DAPI staining and nuclease treatment, strengthening our interpretation that these structures represent membrane lipids involved in telomere anchoring to the nuclear envelope.

This interpretation aligns with the observed data and known nuclear organization. However, we acknowledge that further investigations would be beneficial to conclusively identify these structures.

The revised Figure legend is as follows:

“After digestion with micrococcal nuclease, most BFCs were digested, leaving behind residual dots (indicated by red arrows). Some dots appeared green, unlike the typical blue of DAPI-stained DNA. We interpret these green signals as membrane lipids associated with telomeric anchorage to the nuclear envelope. This interpretation is based on several factors: their resistance to nuclease digestion, which suggests a non-DNA composition; the shift from blue to green fluorescence, which is consistent with DAPI's known interaction with lipids [26]; and their localization, which aligns with the expected position of telomere attachment sites on the nuclear envelope. While further investigation using specific lipid stains and telomere probes would be beneficial to confirm this interpretation, the observed characteristics are consistent with lipid-rich structures.”

Line 196: “ …. DNA dyes, which only visualize condensed DNA ….”

Write a name of this dye.

Response:  

The relevant section has been revised:

“DNA dyes like DAPI, which primarily visualize condensed DNA, revealed bright fluorescent structures representing inner chromosome compartments that remained partially condensed after chelation or processing. In contrast, peripheral chromosome loops (PCLs) [2] appeared as non-fluorescent, birefringent voids, creating 'apparent gaps' between chromosomes.”

Line 288-291 (Figure 3_legend): “….  Figure 3. Additional restriction enzyme(s) pre-digestion led to more resolved higher resilient MNase-resistant fractions. (A) Diagrammatic representation of micrococcal nuclease-mediated BFC disassembly in the absence of restriction enzymes ….”

A more direct expression is more suitable for the figure title to clearly convey its significance.

Example:

High-resolution analysis of micrococcal nuclease-resistant regions reveals DNA-protein interactions and chromosomal anchoring structures.

Response:  

We agree with this comment and have revised the Figure title as suggested.

Figure 3A: Can you please clarify what "component 1-3" refers to?

Response: 

Thanks for pointing this out. The BFCs of dinoflagellates are hypothesized to organize into three distinct compartments [23]:

We proposed Compartment 1 (C(i)) is the highly condensed inner core with a columnar-hexagonal mesophase, exhibiting the highest DNA packaging density. Compartment 2 (C(ii.1)) is the less dense surface layer that spirals around C(i), consisting of chromonema coils. Compartment 3 (C(ii.2)) forms the outermost layer, comprising peripheral chromosomal loops (PCLs) that are transcriptionally active. C(ii.1) and C(ii.2) exist in a dynamic equilibrium, with C(ii.2) capable of condensing into C(ii.1), especially during mitosis. This compartmental organization allows for unique structural flexibility and functional regulation through soft-matter phase transitions, which are crucial for the distinctive chromosome architecture and function in dinoflagellates [23].

To avoid confusion, we have added this information to the revised Figure legend.

Line 306: “2.4. The non-nuclear envelope chromosome ends contain the rDNA loci-NOC that contributed to ….”

Are NOC and NOR the same in meaning? If so, it will standardize the term to NOR.

Response: 

Thanks for pointing out the potential confusion between NOC and NOR (nucleolus organizer regions).

We have standardized the terminology to NOR and hope this clarification addresses your concern.

Line 332: “…. were organized in NOC-like domain, which in other cells ….”

What is NOC-like domain?

Response:  

We have consistently used "NOR" instead of "NOC-like".

Line 492 (Figure 4_legend): “Figure 4. Sirtinol altered telosomal enclaves and cell cycle progression.”

A more direct expression is more suitable for the figure title.

Example:

H3K9 acetylation directly leads to chromosome termini decompaction and cell cycle delayA more direct expression is more suitable for the figure title to clearly convey its significance.

Response:  

Thanks for the suggestion. We have revised the Figure 4 title as follow:

“Sirtinol-mediated chromosome decompaction led to telosomal enclaves expansion within nuclear envelope”

Line 533 (Figure 5_legend): “Figure 5. Chaetocin inhibited H3K9me3, cell proliferation and delayed cell cycle progression.”

A more direct expression is more suitable for the figure title to clearly convey its significance.

Example:

Inhibition of methyltransferase suppresses cell proliferation and cell cycle progression

Response:  

Because there are many methyltransferases, we would suggest a compromise that incorporates the reviewer's concern for more specificity while maintaining scientific accuracy:

Here is the revised title for Figure 5:

“H3K9me3 reduction suppressed exit of S phase and cell proliferation”

Comment 2: While the paper covers various techniques like Comparative Micrococcal Nuclease profiling and the use of chemical agents such as Sirtinol (a promoter histone H3 acetylation and decompaction chromosomal termini), Chaetocin (a histone methyltransferase inhibitor), and Psoralen (a cross-linker supercoiled DNA), it would help to more explicitly explain why these methods were chosen.  The use of chemical agents such as Sirtinol, Chaetocin, and Psoralen may raise concerns about overall toxicity. It might be useful to consider ways to limit their action to specific functions, potentially addressing this issue in the manuscript.

Response 2:

We have revised our manuscript to include the following clarifications:

  1. Choice of techniques:
    • [In section 4.4] Micrococcal nuclease (MNase) resistance profiling is a widely established technique for analyzing nucleosomal structures due to its ability to selectively digest linker DNA while preserving nucleosome-bound DNA fragments [29, 30]. The notably large nuclei of K. brevis (~10 µm) maintain their compacted state and exhibit birefringence in nuclei isolation buffer (NEB) [1], allows us to perform MNase digestion under controlled conditions to assess the dosage-dependence of nuclease-resistant domains in relation to chromosomal architecture.
    • [In section 2.6] Sirtinol, a sirtuin deacetylase inhibitor [15], was employed to investigate the role of histone deacetylation in dinoflagellates chromatin compaction and NE integrity. In various eukaryotic systems, including plant and mammalian cells, sirtinol indirectly promotes H3K9 acetylation, a modification associated with active transcription and open chromatin states [14, 22]. …… In our study of the dinoflagellate C. cohnii, given the significant number of chromosomes and the permanent presence of telomeric ends within the nuclear envelope, we hypothesized that any decompaction resulting from methylation inhibition could potentially affect NE integrity.
    • [In section 2.6] To further elucidate the role of histone modifications in cell proliferation and chromatin structure, we treated Crypthecodinium cohnii with chaetocin, a selective inhibitor of the lysine-specific histone methyltransferase SUV39H1 responsible for H3K9me3 [31]. Despite the absence of architectural nucleosomes in dinoflagellates, chaetocin significantly impeded cell proliferation, ….
    • [In section 2.7] Psoralen, a DNA cross-linking agent that preferentially binds to supercoiled DNA, was utilized to enhance the stability of histone-containing nucleosome complexes during chromatin/nucleosome isolation and electrophoretic analyses [32, 33]. Because the crosslinking occurs in linker DNA, whereas the nucleosomal DNA is protected, which allows to distinguish whether a DNA region had been occupied by a nucleosome or not [33]. Surprisingly, psoralen treatment enhanced the persistence of histone-containing complexes …
  1. Addressing non-specific concerns:
    • We have mentioned that the sirtinol concentration used in our study was lower than those reported in mammalian cell studies. We have revised our comments to reflect that the observed effects of sirtinol could be related to the dual effects of both redox and histone epigenetics, despite the typically low permeability of dinoflagellate amphiesma to extracellular agents.
    • [In section 4.3] In our study, we applied sirtinol at concentrations ranging from 10 to 100 µM, which is 5-fold lower than typically used for mammalian cells [34].  Treatment durations were 3, 6, 9, and 24 hours, an approach designed to minimize prolonged exposure and potential off-target effects. Notably, at earlier time points (e.g. 6 hours) and lower concentrations (10 and 50 µM), the effects of sirtinol were less pronounced, with no observable nuclear swelling. This observation suggests that sirtinol's effects are threshold-dependent, rather than strictly dose-dependent. The gradual onset of observable changes indicates that a certain level of sirtinol accumulation or duration of inhibition may be necessary to induce significant alterations in nuclear structure.
    • [In Discussion] We treated cells with chaetocin at a concentration of 20 µM for 24 hours following cell cycle synchronization. Chaetocin, known to inhibit SUV39H1, not only reduces H3K9me3 levels [35] but may also disrupt redox-sensitive enzymes [36]. While the potential cytotoxic effects of chaetocin are likely related to alterations in cellular redox state, which likely affect higher-order chromosome structure, we observed that cell cycle progression into the G2/M phase was not inhibited. This suggests that the chromatin changes induced by chaetocin treatment do not completely block cell cycle advancement, possibly due to compensatory mechanisms or the threshold of disruption required to halt cell cycle progression.
    • [In Materials and Methods] In vivo crosslinking with psoralen has been widely used for nucleosomes mapping [37]. Our experimental conditions (200 µM psoralen, 30 minutes incubation) were comparable to those used in previous studies on mammalian cells (200 µM) [37] and budding yeast (2 mM, 90 minutes incubation) [38, 39] .

Comment 3: While the study presents novel insights into dinoflagellate chromosomes, expanding on the broader implications of the findings would further enhance its impact. How do these findings contribute to our universal understanding of chromosomal biology in other organisms? What future research directions could stem from these results? Addressing these questions would significantly strengthen the paper's contribution to the field.

I have added further comments to the PDF for your reference, which should assist in the revisions necessary to make this work suitable for publication in IJMS.

Response 3:

We have revised our manuscript to include the following additions at the end of the Discussion:

“Both BfCs and TAs address genome condensation and accessibility challenges in the context of duplication and partitioning, despite dinoflagellates' large genomes and quasi-condensed chromosomes. This offers valuable insights into chromosome engineering across species. The TN configuration provides a unique perspective on DNA packaging without conventional nucleosomes, challenging traditional views on chromatin dynamics and offering new insights into genome organization across diverse organisms. It also highlights the evolutionary adaptability of chromosome architecture and may reveal universal principles in chromosome biology, balancing DNA compaction and accessibility.

The unique TA configuration may necessitate new chromosome isolation techniques to preserve telomere associations, particularly for genome sequencing and epigenetic mapping. Additionally, the telomeric-nuclear envelope insertions likely contribute to the mechanical sensitivity of many dinoflagellate species, which is relevant to their roles as symbiotic zooxanthellae in corals and as regular phytoplankton.“

Reviewer 2 Report

Comments and Suggestions for Authors

The manuscript reports a large amount of experimental and microscopic observations of chromosomal structures in the dinoflagellates Karenia brevis and Crypthecodinium cohnii. What question or hypothesis is being addressed is unclear, however, and accordingly the presentation of the data seems to lack a logical organization. These issues make it difficult to read. The foundation based on which this study was launched, i.e. that dinoflagellates possess telomeric nucleosomes, appears to be assumptive.

Title

The title does not really capture the major finding or conclusion of the work.

Abstract

While the abstract provides a clear conclusion about the telomeric repeats constitute nucleosomal-octameric MNRPs that provides chromosomal anchorage at the nuclear envelope, I do not see convincing results in the Results section that nucleosomes are found in the NE-attached chromosome end.

Introduction

Telomere: it is important to point out, with references, that the plant-like telomere repeats have been found in multiple species and from genomes as well as other types of data. Data to date show that dinoflagellate tolemeric sequence is identical although length might be variable but is not clear.

Lines 43-44: is there any published data supporting this statement? Without data, it should be toned down to reflect that it is your speculation.

Lines 57-58: these two complement each other instead of contrast.

Line 61: by “differs” do you mean euchromatin is organized inside and heterochromatin is outside? Provide the explanation in the manuscript.

Lines 65-92: indicate, wherever applicable, what organisms this information is from.

Lines 90-92: Not sure what made the authors believe dinoflagellates have telomeric nucleosomes. At the very least, references should be provided to support the claim.

Results

Generally, this section contains much discussion of the results, with unclear wording as to whether previously published results from other eukaryotes are described or new findings from the current study are reported.

How was H2B antibody produced and purified?

Fig. S2A: how come GAPDH was concentrated in the insoluble fraction?

Fig. 2: “resistant” looks to me more like “incomplete” due to insufficient time given.

Line 247: I do not think “resilient” is the correct wording; it was “protected”

Fig. 3E: “deprotienated” a typo?

Lines 321-322: is there evidence that rDNAs are located at chromosome ends?

Line 360-361: it can be NE at a deeper plane, as another explanation.

Line 370: sirtinol has two functions: iron chelator and an inhibitor of the deacetylase sirtuin. Which role did you use it for? Explain it in the paper.

Lines 377-379: ref [40] is about Drosophila, which should be specified. In addition, the paper indicates that myosin is part of the nuclear pore. This is different to say “around the critical domain adjacent to the NE”, and I do not see how it implies “that telomeric nucleosomes are involved in chromosome mobility…” Nuclear pores mainly facilitates RNA export and nuclear protein import.

Lines 396-398: The references cited here indicate that rDNA (e.g. 5S rDNA) spread in chromosomes like mobile elements, that rDNA is associated with other mobile elements.

Lines 450-464: it is problematic to compare dino histone H3 with human H3 to infer distinct feature when because these organisms are so far apart. It would be more appropriate to compare dinos with perkinsids or apixomplexa or diatoms. To infer unique feature, it is best to identify common and conserved features across a broad range of organisms. Besides, I do not understand the statement “…likely evolved with the loss of canonical core histones…” while describing H3 in the species. Isn’t H3 one of the core histones? Fig. S2 shows the detection of H2A, and Fig. S4 as well as Fig. 6 show presence of H3.

Fig. 6: are these denaturing gels? The molecular masses suggest the proteins were in complexes, but the figure legend is not clear.

Lines 467-488: explicit wording is needed to indicate what is previously reported findings in what organisms, and what is findings or speculation of the current study about the dinoflagellate species.  

Discussion

Because a lot of discussion is already given in the Results section, the discussion section does not add much. I would suggest that Results and Discussion be combined. However, the materials need to be much better organized to make it easier to follow.

Comments on the Quality of English Language

English language in individual sentences is fine in general, but the organization of the manuscript is not easy to follow.

Author Response

Author's Reply to the Review Report (Reviewer 2)

Point-by-point response to Comments and Suggestions for Authors

The manuscript reports a large amount of experimental and microscopic observations of chromosomal structures in the dinoflagellates Karenia brevis and Crypthecodinium cohnii. What question or hypothesis is being addressed is unclear, however, and accordingly the presentation of the data seems to lack a logical organization. These issues make it difficult to read. The foundation based on which this study was launched, i.e. that dinoflagellates possess telomeric nucleosomes, appears to be assumptive.

Response:

Thank you for your thoughtful feedback on our manuscript.

In response to the comments, we have made the following revisions:

  • To address concerns about the paper's length and complexity, particularly in the Results section, we have carefully rewritten the relevant sections (especially those highlighted) by using more concise language and reorganized the content in a more logical order to condense and simplify the explanations
  • We have shortened, reorganized and rewritten the whole Discussion section to improve its flow and clarity
  • We have added some explanatory notes to several figure legends (e.g. Figures 3A, 6, and S4) to clarify specific terms and key points
  • We have revised some Figure title, to better coney the significance of the Figure

We hope these changes would significantly improve the accessibility of our work.

Comment 1:

Abstract

While the abstract provides a clear conclusion about the telomeric repeats constitute nucleosomal-octameric MNRPs that provides chromosomal anchorage at the nuclear envelope, I do not see convincing results in the Results section that nucleosomes are found in the NE-attached chromosome end.

Response 1:

Regarding the evidence that support the presence of nucleosomes at the nuclear envelope (NE)-attached chromosome ends in dinoflagellates, we would like to provide further clarification and highlight specific results from our study that substantiate this claim.

  1. Selective Resistance Patterns: The absence of significant effects from sirtinol on the bulk of Birefringent Chromosomes (BFCs) at lower concentrations (Figure 4) indicates that the majority of nucleosomal structures are localized to specific regions rather than being uniformly distributed across all chromosomes. This selective resistance suggests a specialized chromatin organization at telomeric regions.
  2. Distinct Mobility on Two-Dimensional Gels: Our two-dimensional gel analyses (Figure 7) revealed distinct mobility patterns in MNase-digested, psoralen-crosslinked, and non-crosslinked nuclei samples. The different slopes observed with telomeric probes indicate that telomeric nucleosomes (TNs) possess unique properties, such as higher mobility and lower superhelicity, distinguishing them from canonical nucleosomes. This distinct behavior is consistent with the presence of specialized nucleosomal structures at telomeric regions.
  3. Absence of Nucleosomal Ladder with cDNA Probes: The lack of a typical nucleosomal ladder pattern when using cDNA hybridization probes (Figure S3B) further supports the conclusion that nucleosomes have less stability than expected, based on the pI of dinoflagellate core histones, and localized to telomeric regions rather than being distributed along gene-encoding domains. This specificity reinforces the notion that TNs are the primary nucleosomal structures in dinoflagellates.
  4. Histone Variant Localization: Our immunofluorescence data demonstrated that H3K9me3, including that on the potential telomere-enriched H3.3, at the NE-attached chromosome ends (Figure 4A). The enrichment of these histone variants, which are known to form more dynamic and less stable nucleosomes, underscores their role in facilitating chromatin accessibility and maintenance at telomeric regions.
  5. Immobility of BFCs and NE Insertions: Transmission electron microscopy (TEM) photomicrographs (Figure 1D) and observations from time-lapse videography (Video S1) show the immobility of BFCs and the presence of nuclear envelope insertions. These insertions likely represent telomeric nucleosome anchorages, providing a physical link between chromosome ends and the NE.
  6. Biochemical Interactions: The combination of MNase-resistant profiles and psoralen cross-linking experiments (Figure 7) suggests that TNs are stabilized through protein-DNA interactions, further supporting their role in anchoring chromosomes to the NE.

Collectively, these findings provide evidence that telomeric nucleosomes are integral to the anchorage of chromosome ends at the nuclear envelope in dinoflagellates. We have incorporated these clarifications and references into the manuscript to ensure the robustness of our conclusions.

We have also revised the Abstract as follows:

“Dinoflagellate Birefringent chromosomes (BfCs) contain some of largest known genomes yet lack typical nucleosomal micrococcal-nuclease protection pattern, despite containing variant core histones. One BfC end interacts with extranuclear mitotic microtubules at the nuclear envelope (NE), which remains intact throughout the cell cycle. Ultrastructural studies, polarized light and fluorescence microscopy, and micrococcal nuclease-resistant profiles (MNRPs), revealed that NE-associated chromosome ends persisted post-mitosis. Histone H3K9me3 inhibition caused S-G2 delay in synchronous cells, without any effects at G1. Differential labeling and nuclear envelope swelling upon decompaction indicate an extension of the inner compartment into telosomal anchorages (TAs). Additionally, limited effects of low-concentration sirtinol on bulk BfCs, coupled with distinct mobility patterns in MNase-digested and psoralen-crosslinked nuclei observed on 2D gels, suggest that telomeric nucleosomes (TNs) are the primary histone structures. The absence of a nucleosomal ladder with cDNA probes, the presence of histone H2A and telomere-enriched H3.3 variants, along with the immuno-localization of H3 variants mainly at the NE further reinforce telomeric regions as the main nucleosomal domains. Cumulative biochemical and molecular analyses suggest that telomeric repeats constituted the major octameric MNRPs that provision chromosomal anchorage at the NE.”

Comment 2:

Introduction

Telomere: it is important to point out, with references, that the plant-like telomere repeats have been found in multiple species and from genomes as well as other types of data. Data to date show that dinoflagellate tolemeric sequence is identical although length might be variable but is not clear.

Response:

We have revised the paragraph (line 35-42) to include the following additional information:

"Historically, the structure of dinoflagellate chromosomes was debated, with earlier models proposing a circular architecture [40]. However, contemporary evidence, including telomerase activity demonstrated via the Telomeric Repeats Amplification Protocol (TRAP) assays [7], and the successful application of peptide nucleic acid (PNA) probes in in situ hybridization experiments, has confirmed that dinoflagellate chromosomes are linear and capped with plant telomeric sequences (TTTAGGG)n [14]. It is important to note that this plant-like telomere repeat sequence has been consistently found in evolutionarily distant dinoflagellate species (which also varies in their chromosome size and DNA content), including athecate species like Karlodinium veneficum [41], Karenia brevis [41], Amphidinium carterae [42] and thecate species like Alexandrium minutum [43], Prorocentrum micans [42], as evidenced by both genomic (satellitome) [43, 44] and FISH (fluorescence in situ hybridization) [7, 41, 42, 45, 46] data.  While the sequence itself appears to be identical across studied dinoflagellate species, the length of the telomeric repeats may vary. Notably, pulse-field gel electrophoresis of the Bal 31-digested Karenia brevis nuclei has shown that these telomeric DNA lengths are longer than commonly observed for other protists [7]. However, the exact extent of length variation among different dinoflagellate species remains to be fully elucidated.”

Thank you for your valuable feedback, which helps improve the accuracy and depth of our manuscript.

Lines 43-44: is there any published data supporting this statement? Without data, it should be toned down to reflect that it is your speculation.

Response:

Thanks for bringing this to our attention. We have deleted the statement.

Lines 57-58: these two complement each other instead of contrast.

Response:

For the statement: “Previous in-gel restriction enzyme digestion experiments demonstrated the release of gene-encoding domains [5], which reside in the outer compartment of dinochromosomes, known as the peripheral chromosomal loops (PCLs). This contrasts with the conceptual inner "structural DNA" proposed by Sigee et al. (1986) [2].”

We agree that these two concepts complement each other rather than contrast.

The relevant section is revised as follows:

"Previous in-gel restriction enzyme digestion experiments demonstrated the release of gene-encoding domains [5], which resided in the outer compartment of dinochromosomes, known as the peripheral chromosomal loops (PCLs). This finding complements the concept of inner "structural DNA” [2], suggesting a complex, multi-compartmental organization of dinochromosomes."

Line 61: by “differs” do you mean euchromatin is organized inside and heterochromatin is outside? Provide the explanation in the manuscript.

Response:

We agree that our use of the word "differs" requires further clarification. We did not mean to imply that euchromatin is organized inside and heterochromatin outside in dinoflagellates. Rather, we intended to highlight the unique organization of dinoflagellate chromosomes compared to the typical eukaryotic model.

To address the comment and provide a clearer explanation, we revised this section as follows:

"The inner-outer compartmentation of dinoflagellate chromosomes exhibits some conceptual parallels with the heterochromatin-euchromatin distinction observed in other eukaryotes [47]. However, the organization in dinoflagellates is unique: the outer compartment (peripheral chromosomal loops or PCLs) contains the only actively transcribed genes, while the inner core likely contains no transcriptionally active domains and is termed structural DNA, effectively separating the coding and non-coding sequences. This arrangement differs from typical eukaryotes, where euchromatin and heterochromatin are often interspersed with both coding and non-coding sequences throughout the chromosome. In dinoflagellates, this clear spatial separation, with transcriptionally active regions exclusively in the outer compartment, represents a distinct chromosomal architecture. This unique division of labor likely employs subcompartmentation for differential gene expression, facilitating the multiple life-cycle stage transitions characteristic of dinoflagellates."

This revision provides a more detailed explanation of how the compartmentation in dinoflagellate chromosomes compares to and differs from the heterochromatin-euchromatin organization in typical eukaryotes. It clarifies that the uniqueness lies in the clear spatial separation of transcriptionally active regions to the outer compartment, rather than implying an inverted organization of euchromatin and heterochromatin.

We believe this explanation addresses the question and provides the necessary context for readers to understand the unique aspects of dinoflagellate chromosome organization.

Lines 65-92: indicate, wherever applicable, what organisms this information is from.

Response:

The information is based on plant and mammalian cells studies. Here's the revised paragraphs with this information included:

" Telomeres, located at the ends of chromosomes in eukaryotes, consist of repetitive non-coding DNA sequences. In both plant and mammalian cells, special mechanisms are installed to ensure telomere from progressively shortening during each round of DNA replication, as the 5' end of the lagging strand cannot be synthesized after the removal of the last RNA primer, resulting in a 3' overhang [48-50]. Telomeres and telomeric nucleosomes (TNs) are widely acknowledged as vital protective structures that safeguard the integrity of chromosome ends during replication [50, 51]. However, their potential roles in other cellular processes are less addressed.

Although canonical histone complements are significantly reduced in dinoflagellate genomes, histone variant presence suggests essential roles in chromosomal transactions that are perhaps commonly obscured by canonical architectural nucleosomes in other organisms. In both plant and mammalian system, histone H3 at lysine 9 methylation (including H3K9me3 and H3K9me2) is a critical epigenetic marker that influences the compaction state of telomeric nucleosomes and rRNA loci [52-56]. Its interaction with the Su(var)3-9 histone H3-K9 methyltransferase (Suv39H, plant homologue NtSET1 and SUVH4/KYP) stabilizes telomeric nucleosomes, with Suv39H null mutants showing decreased H3K9me3 levels and abnormal telomere elongation, demonstrating the importance of H3K9me3 in maintaining telomeric integrity [57-60]. The retention of telomeric nucleosomes (TNs) in dinoflagellates, despite the absence of canonical architectural nucleosomes, implicates their potential roles in orchestrating system-level chromosomal organization in these unique protists.”

Lines 90-92: Not sure what made the authors believe dinoflagellates have telomeric nucleosomes. At the very least, references should be provided to support the claim.

Response:

Thank you for the insightful feedback on our manuscript. Regarding the evidence that support the presence of nucleosomes at the nuclear envelope (NE)-attached chromosome ends in dinoflagellates, we would like to provide further clarification and highlight specific results from our study that substantiate this claim. We agree that the immunocaptured complex, showing reciprocal immuno-positive signals with anti-H3 and anti-H2B antibodies (Figure 6A), suggesting an octameric rather than a nucleosomal definition. We have accordingly rephrased this to "octameric" to reflect the higher molecular weight.

  1. Selective Resistance Patterns: We observed highly MNRPs manifesting after extended MNase treatment, rather than appearing early with lesser resistance. This, coupled with NE-located condensed DNAs after MNase digestion and the shifted fluorescence of SYTOX green, reasonably suggests that the TA located chromosome ends are enriched with MNase-resistant histone higher-order structures (HHOS) that corresponds with the higher dynamics of telomeric nucleosomes in other systems.
  2. Distinct Mobility on Two-Dimensional Gels: Our two-dimensional gel analyses (Figure 7) revealed distinct mobility patterns in MNase-digested, psoralen-crosslinked, and non-crosslinked nuclei samples. The different slopes observed with telomeric probes indicate that telomeric nucleosomes (TNs) possess unique properties, such as higher mobility and lower superhelicity, distinguishing them from canonical nucleosomes.  This, coupled with the histone-mediated complex observed after MNRPs, logically suggests the presence of higher-order histone-DNA structures conducive to the second slopes observed in 2-D gel electrophoresis.
  3. Histone Variant Localization: Our immunofluorescence data demonstrated that H3K9me3, including that on the potential telomere-enriched H3.3, at the NE-attached chromosome ends (Figure 4A). The enrichment of these histone variants, which are known to form more dynamic and less stable nucleosomes, underscores their role in facilitating chromatin accessibility and maintenance at telomeric regions.
  4. Biochemical Interactions: The combination of MNase-resistant profiles and psoralen cross-linking experiments (Figure 7) suggests that TNs are stabilized through protein-DNA interactions, further supporting their role in anchoring chromosomes to the NE.

Collectively, these findings provide evidence that telomeric nucleosomes are integral to the anchorage of chromosome ends at the nuclear envelope in dinoflagellates. We have incorporated these clarifications and references into the manuscript to ensure the robustness of our conclusions.

We have also revised the Abstract as follows:

“Dinoflagellate Birefringent chromosomes (BFCs) contain some of largest known genomes yet lack typical nucleosomal micrococcal-nuclease protection pattern despite containing variant core histones. One BFC end interacts with extranuclear mitotic microtubules at the nuclear envelope (NE), which remains intact throughout the cell cycle. Ultrastructural studies, polarized light and fluorescence microscopy, and micrococcal nuclease-resistant profiles (MNRPs), revealed that NE-associated chromosome ends persisted post-mitosis. Histone H3K9me3 inhibition caused S-G2 delay in synchronous cells, without any effects at G1. Differential labeling and nuclear envelope swelling upon decompaction indicate an extension of the inner compartment into telosomal anchorages (TAs). Additionally, limited effects of low-concentration sirtinol on bulk BFCs, coupled with distinct mobility patterns in MNase-digested and psoralen-crosslinked nuclei observed on 2D gels, suggest that telomeric nucleosomes (TNs) are the primary histone structures. The absence of a nucleosomal ladder with cDNA probes, the presence of histone H2A and telomere-enriched H3.3 variants, along with the immuno-localization of H3 variants mainly at the NE further reinforce telomeric regions as the main nucleosomal domains. Cumulative biochemical and molecular analyses suggest that telomeric repeats constituted the major nucleosomal-octameric MNRPs that provision chromosomal anchorage at the NE. “

Comment 3:

Results

Generally, this section contains much discussion of the results, with unclear wording as to whether previously published results from other eukaryotes are described or new findings from the current study are reported.

Response:

In response to the comments, we have made the following revisions:

  • To address concerns about the paper's length and complexity, particularly in the Results section, we have carefully rewritten the relevant sections (especially those highlighted) by using more concise language and reorganized the content in a more logical order to condense and simplify the explanations
  • We have shortened, reorganized and rewritten the whole Discussion section to improve its flow and clarity
  • We have added some explanatory notes to several figure legends (e.g., Figures 3A, 6, and S4) to clarify specific terms and key points
  • We have revised some Figure title, to better coney the significance of the Figure

We hope these changes would significantly improve the accessibility of our work.

How was H2B antibody produced and purified?

Response:

We revised the relevant section in our Materials and Methods as follows:

"Polyclonal anti-dHlp antibody was generated against full-length bacterial expressed CcHCc3p and should have recognized other monomeric dHLP isotypes that exhibit high homology and similar sizes (13-14 kDa) [61]. Anti-histone H2B (sc-10808), anti-histone H3 (sc-8645R), and anti-α-tubulin (sc-53646) antibodies were purchased from Santa Cruz. All the HCc3 (HLP), H2B and H3 antibodies were affinity-purified with membrane-bound antigens (Crypthecodinium cohnii HCc3, H2B and H3) [62] after pre-cleaning with dried bacterial acetone powder [63]. Cell lysate preparation and immunoblotting essentially followed published protocols [64, 65]. Band intensities were determined using ImageJ software (NIH) [66]. All PCR primers were listed in Table S1."

Fig. S2A: how come GAPDH was concentrated in the insoluble fraction?

Response:

The result was now deleted

Fig. 2: “resistant” looks to me more like “incomplete” due to insufficient time given.

Response:

We understand the concern that "resistant" might imply incomplete digestion due to insufficient time. However, we would like to clarify that our MNase digestion experiments were conducted for extended periods, up to 16 or 24 hours, as indicated in the figure. These extended digestion times are generally considered more than sufficient for complete digestion of accessible chromatin in most experimental systems.

Given these extended digestion times, we believe that the persistence of certain DNA fragments is indeed indicative of resistance to MNase digestion, rather than incomplete digestion. This resistance likely reflects structural features or protein associations that protect these DNA regions from enzymatic cleavage, even over prolonged periods.

However, we acknowledge that the term "resistant" could be further clarified to avoid any potential misinterpretation. We revised our description as follows:

"MNase-protected domains, revealed by overnight digestion (24 hours) and probed with rDNA, indicated the presence of persistently protected rDNA loci."

Line 247: I do not think “resilient” is the correct wording; it was “protected”

Response:

We agree that "protected" is a more appropriate term in this context, and this change would better reflect the nature of these chromatin domains and align with standard terminology in the field.

We revised the paragraph as follows:

"Isolated chromosomes did not exhibit prominent MNase-protected domains that whole nuclei did (red box), with apparent m.w. lower than 100 bp, and the protected domain remained largely undigested even after 24 hours. The MNRP Southern blot (Figure 2C-F) suggested that this fraction was neither telomeric nor contained rDNA. The highly protected fraction (HPF), retained within the nucleus and ranging between 3-12 kbp (Figure 2E, highlighted by the green box), showed only a weak Southern blot signal with rDNA probes, indicating the presence of major protected structures in addition to the non-HPF rDNA loci."

Fig. 3E: “deprotienated” a typo?

Response:

Thank you for pointing out this typo error. Fixed.

Lines 321-322: is there evidence that rDNAs are located at chromosome ends?

Response:

  1. Previous TEM studies: As mentioned in our manuscript, previous TEM studies have indicated that nucleoli consist of decondensed chromosomal loci [4]. This suggests that rDNA, which is typically associated with nucleoli, is likely located at chromosomal regions that can decondense into the nuclear center.
  2. Southern blot analysis: Our Southern blot using rDNA probes showed a high molecular weight, MNase-resistant fraction in the early time points (Figure 2F, red squares). This suggests that rDNA is present in regions that are relatively protected from MNase digestion, which is consistent with their potential localization at chromosome ends.
  3. Metripol and birefringence microscopy: Our observations suggested that the non-NE (Figure 2A, B) ends of chromosomes are 'opened' into the nuclear center where nucleoli are located. This is consistent with rDNA being positioned at these open ends.
  4. Absence in isolated chromosomes: The high molecular weight rDNA-positive bands were notably absent from the MNase-resistant protected patterns (MNRPs) of isolated chromosomes. This supports the hypothesis that the B termini (non-NE ends) were cleaved during chromosome isolation, with these open ends being the likely location(s) of rDNA repeats.
  5. Lack of association with telomeric repeats: The absence of association between rDNA-positive signals and telomeric repeat signals at extended digestion times suggests that rDNAs are not present at the NE end (which is associated with telomeres), but rather at the opposite end.

These points collectively suggest that rDNAs are likely located at the non-NE chromosome ends.

Line 360-361: it can be NE at a deeper plane, as another explanation.

Response:

Metripol orientation analysis reported on the sequential changes in anisotropic orientation from focused circular polarized light at the layer-specific response, rather than from deeper layers.

Line 370: sirtinol has two functions: iron chelator and an inhibitor of the deacetylase sirtuin. Which role did you use it for? Explain it in the paper.

Response:

We have mentioned that the sirtinol concentration used in our study was lower than those reported in mammalian cell studies. The observed effects of sirtinol could be related to the dual effects of both redox and histone epigenetics, despite the typically low permeability of dinoflagellate amphiesma to extracellular agents.

[In section 4.3] In our study, we applied sirtinol at concentrations ranging from 10 to 100 µM, which is 5-fold lower than typically used for mammalian cells [34].  Treatment durations were 3, 6, 9, and 24 hours, an approach designed to minimize prolonged exposure and potential off-target effects. Notably, at earlier time points (e.g. 6 hours) and lower concentrations (10 and 50 µM), the effects of sirtinol were less pronounced, with no observable nuclear swelling. This observation suggests that sirtinol's effects are threshold-dependent, rather than strictly dose-dependent. The gradual onset of observable changes indicates that a certain level of sirtinol accumulation or duration of inhibition may be necessary to induce significant alterations in nuclear structure.

Lines 377-379: ref [40] is about Drosophila, which should be specified. In addition, the paper indicates that myosin is part of the nuclear pore. This is different to say “around the critical domain adjacent to the NE”, and I do not see how it implies “that telomeric nucleosomes are involved in chromosome mobility…” Nuclear pores mainly facilitates RNA export and nuclear protein import.

Response:

We have decided to remove this paragraph entirely.

Lines 396-398: The references cited here indicate that rDNA (e.g. 5S rDNA) spread in chromosomes like mobile elements, that rDNA is associated with other mobile elements.

Response:

The section has been revised:

“Repetitive elements, including those associated with rDNAs, play a significant role in genomic evolution and organization. Previous studies have shown that rDNAs, such as 5S rDNA, can spread throughout chromosomes in a manner similar to mobile elements  [67-69]. This distribution of repetitive elements, including those associated with rDNAs, often marks major evolutionary events and contributes to genomic diversity.”

Lines 450-464: it is problematic to compare dino histone H3 with human H3 to infer distinct feature when because these organisms are so far apart. It would be more appropriate to compare dinos with perkinsids or apixomplexa or diatoms. To infer unique feature, it is best to identify common and conserved features across a broad range of organisms. Besides, I do not understand the statement “…likely evolved with the loss of canonical core histones…” while describing H3 in the species. Isn’t H3 one of the core histones? Fig. S2 shows the detection of H2A, and Fig. S4 as well as Fig. 6 show presence of H3.

Response:

We have aligned the H3 variants in C. cohnii with canonical H3 proteins from more closely related organisms, including apicomplexans, perkinsids, and diatoms (the latter two having H3 but not H3.3). This comparison revealed that C. cohnii H3 variants exhibit unique features distinct from both their close relatives and more distant eukaryotes. Specifically, they show regions with highly variable sequences compared to apicomplexans and possess extra sequences in both the N-terminal and histone fold domains not found in other eukaryotes.

Here is the revised version of the Results section, based on the updated Figure S4:

"Histone H3.3 is a variant of H3, differing from the canonical H3 by only a few amino acids in evolutionarily distinct organisms including humans, Drosophila, Xenopus, and Arabidopsis [70, 71].

Multiple sequence alignment of C. cohnii H3 variants with canonical H3 proteins from other organisms, including apicomplexans, perkinsids, and diatoms (the latter two having H3 but not H3.3), revealed that C. cohnii H3 variants exhibit unique features distinct from both their close relatives and more distant eukaryotes (Figure S4). C. cohnii H3 variants show regions with highly variable sequences compared to apicomplexans and possess extra sequences in both the N-terminal and histone fold domains not found in other eukaryotes.

Moreover, C. cohnii H3 variants do not share the "(A/Q/E/T/H)A(I/L/V)(L/G)" sequence motif conserved in many eukaryotic H3.3 proteins, nor do they possess the "SAV(M/L/A)" motif characteristic of canonical H3 in many organisms. Instead, these variants display a diverse "(Q/E/S)(A/G)(I/L)(L/S/E)" motif, which more closely resembles the H3.3 "(A/Q/E/T/H)A(I/L/V)(L/G)" motif (Figure S4). Despite these sequence differences, both CcH2A and CcH3 variants were still recognizable by commercially available H3 and H2A antibodies used in this study (Figures S1, S2, and Figure 6), which target the highly conserved histone fold region shared by canonical histones and histone variants.

Furthermore, most (5 out of 6) of these C. cohnii variants exhibit similar transcription levels in G1 and S-G2 cells (data from in-house cell cycle transcriptome), reminiscent of the expression pattern of H3.3 in other eukaryotes [11-13].”

And the revised Figure legend for Figure S4, which contains more details:

“Multiple sequence alignment of C. cohnii H3 variants with canonical H3 proteins from other organisms, including apicomplexans, perkinsids, and diatoms (the latter two having H3 but not H3.3), revealed that C. cohnii H3 variants exhibit unique features distinct from both their close relatives and more distant eukaryotes (Figure S4). C. cohnii H3 variants show regions with highly variable sequences compared to apicomplexans and possess extra sequences in both the N-terminal and histone fold domains not found in other eukaryotes. Moreover, C. cohnii H3 variants do not share the "(A/Q/E/T/H)A(I/L/V)(L/G)" sequence motif conserved in many eukaryotic H3.3 proteins, nor do they possess the "SAV(M/L/A)" motif characteristic of canonical H3 in many organisms. Instead, these variants display a diverse "(Q/E/S)(A/G)(I/L)(L/S/E)" motif, which more closely resembles the H3.3 "(A/Q/E/T/H)A(I/L/V)(L/G)" motif (Figure S4). Despite these sequence differences, both CcH2A and CcH3 variants were still recognizable by commercially available H3 and H2A antibodies used in this study (Figures S1, S2, and Figure 6), which target the highly conserved histone fold region shared by canonical histones and histone variants.”

Fig. 6: are these denaturing gels? The molecular masses suggest the proteins were in complexes, but the figure legend is not clear.

Response:

Yes, Figure 6A and 6B are denaturing gels. We have revised the figure legend accordingly:

"…  …  histone H3 in the same immunoprecipitants as (A). Samples were analyzed on denaturing SDS-PAGE gels. The use of psoralen coupled with DSG-crosslinking, when compared to DSG-formaldehyde crosslinking alone, was more efficient in preserving high molecular weight complexes (yellow boxes) that were absent in the mock control. These high molecular weight bands represent crosslinked protein complexes that were not fully dissociated under the denaturing conditions. Lower molecular weight bands/complexes (red boxes), likely corresponding to H2A-H2B dimers or H3-H4 tetramers, were also detected in the IP product. …”

Lines 467-488: explicit wording is needed to indicate what is previously reported findings in what organisms, and what is findings or speculation of the current study about the dinoflagellate species.  

Response:

We have revised the paragraph to explicitly differentiate between established knowledge from other organisms and our novel findings in dinoflagellates.

“Sirtinol, a sirtuin deacetylase inhibitor [15], was employed to investigate the role of histone deacetylation in dinoflagellates chromatin compaction and NE integrity. In various eukaryotic systems, including plant and mammalian cells, sirtinol indirectly promotes H3K9 acetylation, a modification associated with active transcription and open chromatin states [14, 22]. While sirtinol does not directly alter histone methylation, its enhancement of acetylation can indirectly reduce methylation at the same residue. Surprisingly, despite the fewer number of post-translational modifications (PTMs) on C. cohnii histone H3 variants (data not shown), these two marks are well conserved (Figure S4).

Our observations revealed that, unlike AMSA, which dose-dependently caused decompaction of the main body of chromosomes leading to nuclear eruption, the sirtinol effect in C. cohnii was much slower and more regional. We observed that central nucleolar positions became more decondensed prior to the cortical region. This difference in decondensation was evident at 10 and 50 µM concentrations at the 6-hour mark (Figure 4B, C). These findings suggest that sirtinol-induced effects, whether through specific or non-specific effects, affected chromosome higher-order organization at the cortical areas, transmitting to decondensation at the chromosome ends within the NE. Histone acetylation selectively affects chromatin organization near the nucleolus and telomeric regions, thereby influencing NE stability and overall nuclear architecture.

Our experiments also demonstrated dose-dependent restricted BfC decompaction in C. cohnii, without significant nuclear decompaction until reaching 50 µM (Figure 4B). Notably, we found that the eventual decompaction of the main body coincided with DNA staining within the nuclear envelope, which was previously unstainable. This observation unequivocally demonstrated the presence of BfC termini within NE DNAs in C. cohnii, which we refer to as telosomal enclaves for the purpose of discussion, which were connected to the decompaction of telomeric nucleosomes. Based on our findings in C. cohnii, we suggest that despite whole chromosome connection, most of the telomeric nucleosome-mediated compaction occurs in the TA direction, with lesser contribution to the chromosome bulk, which was more associated with topoisomerase.”

Comment 4:

 Discussion

Because a lot of discussion is already given in the Results section, the discussion section does not add much. I would suggest that Results and Discussion be combined. However, the materials need to be much better organized to make it easier to follow.

Response:

We appreciate your suggestion to combine the Results and Discussion sections. However, we have decided to maintain separate Results and Discussion sections in accordance with the IJMS journal format. We acknowledge your point about the need for better organization, and we have thoroughly revised our Discussion section, added clear headings and strengthened the connections between ideas. These changes aim to significantly improve the section's structure and overall clarity, making it easier for readers to follow our arguments and interpretations.

Please refer to the revised Discussion section.

Response to Comments on the Quality of English Language

Point 1: English language in individual sentences is fine in general, but the organization of the manuscript is not easy to follow.

Response:

  • To address concerns about the paper's length and complexity, particularly in the Results section, we have carefully rewritten the relevant sections (especially those highlighted) by using more concise language and reorganized the content in a more logical order to condense and simplify the explanations
  • We have shortened, reorganized and rewritten the whole Discussion section to improve its flow and clarity
  • We have added some explanatory notes to several figure legends (e.g., Figures 3A, 6, and S4) to clarify specific terms and key points
  • We have revised some Figure title, to better coney the significance of the Figure

We hope these changes would significantly improve the accessibility of our work.

Round 2

Reviewer 2 Report

Comments and Suggestions for Authors

Responses to my previous comments are acceptable. The only other thing is the omission of major mainstream literature when dinoflagellates and their genomic architecture are introduced. I strongly recommend to be up to date. 

Author Response

Dear Reviewer,

Thank you for the comment regarding the inclusion of major mainstream literature on dinoflagellate genomic architecture.

We have added two new paragraphs in the Introduction (please refer to the blue highlighted text in Introduction) that provide a comprehensive overview of dinoflagellate biology, their unique genomic features, and the various models proposed to explain their chromosome structure. Specifically:

The first added paragraph introduces dinoflagellates, their ecological significance, and their distinctive genomic characteristics. We have included up-to-date references on their genome size range, the nature of their chromosomes (BfCs), gene arrangement, and the liquid crystalline state of their DNA.

The second added paragraph discusses several key models proposed to explain dinoflagellate chromosome organization. We have included the "toroidal chromonema" model, the "stacks-of-DNA-discs" model, etc. This addition provides a balanced view of the current understanding and ongoing debates in the field.

These additions draw from recent and seminal literature in the field, ensuring that our introduction is up-to-date and reflective of the current state of knowledge in dinoflagellate genomics.

We believe these revisions address your concern about the omission of major mainstream literature and provide a more comprehensive introduction to dinoflagellate genomic architecture. We thank you for your valuable feedback, which has helped improve the quality and completeness of our manuscript.

Here are the additional paragraphs (please refer to the blue highlighted text in Introduction):

“Dinoflagellates are a diverse and ecologically significant group of unicellular eukaryotic protists within the Alveolata clade, which also includes ciliates and apicomplexans [1]. With approximately 2,500 extant species spanning around 300 genera [2], dinoflagellates play crucial roles as primary marine producers, contributors to harmful algal blooms (HABs), and essential symbionts of reef-building corals. Dinoflagellates exhibit some of the most unusual and complex chromosome and genome structures known among eukaryotes [3]. One of the most remarkable features of dinoflagellates is their exceptionally large genomes, ranging from 1.5 to 250 gigabases [4-6], which are housed within birefringent quasi-condensed chromosomes (BfCs) [7] that lack the canonical histone nucleosomal architecture [8-16]. This unique genome architecture is further marked by unidirectionally arranged genes, often presented in tandem repeats [17-19], and a substantial proportion of transcriptionally active DNA organized into peripheral loops [20]. Additionally, dinoflagellate chromosomes remain permanently condensed throughout the cell cycle, maintaining a highly organized, cholesteric liquid crystalline state with a constant left-handed twist [7, 21]. “

and

“Several models have been proposed to elucidate the intricate organization of dinoflagellate chromosomes. Notable among these are the “toroidal chromonema” model [14], which envisions chromosomes as toroidal bundles of DNA strands based on circular chromosome organization, and the "stacks-of-DNA-discs" (or cholesteric liquid crystal) model [7, 57, 58], which proposed a linear chromosome organization with each chromosome arranged as layered discs of nested DNA arches. Previous models of BfCs, based on purported higher-order coiling, likely represented partially decompacted chromosomes, as would be expected from TEM processing of the central compartment (Figure 1D). It would not be conceptually helpful to evoke 'Liquid Crystalline Chromosomes' to describe the whole chromosomes, as rigidity would be required to sustain the chromosome shape. Comparative TEM studies by the pioneer TEM developer Kellenberger group demonstrated that low-protein nucleic acids (unlike nucleosomal nucleic acid) exhibited apparently artifactual cholesteric 'DNAplasm' [59-61]. Furthermore, recent studies suggest that the surface peripheral chromosome loop domains are transcription-mediated and would not be in liquid crystalline form [62]. Given these considerations, we prefer to refer to the readout description of 'Birefringent Chromosomes'.”